# USER-CONTROLLABLE DENSE VIDEO CAPTIONING: A LARGE-SCALE BENCHMARK AND FRAMEWORK

## ABSTRACT

Dense video captioning (DVC) aims to generate temporally localized captions for multiple events in untrimmed videos. Despite recent advances, existing methods still generate fixed captions because existing benchmarks provide only single-style annotations and methods for handling variations in event granularity and caption specificity remain unexplored. To address this gap, we present User-Controllable Captions (UC Captions), a new dataset with annotations that vary in event density (*i.e.*, how frequently events are detected) and caption depth (*i.e.*, the level of descriptive detail for a given event). This dataset is the first in DVC to explicitly encode controllable dimensions of annotation, establishing a foundation for studying user-driven flexibility. Building on this, we propose User-Controllable DVC (UC-DVC), a framework that incorporates user-defined density and depth parameters to dynamically adjust event localization and caption generation. Extensive experiments demonstrate that UC-DVC flexibly adapts to diverse user requirements while maintaining competitive performance on standard benchmarks. To support further research, both UC Captions and UC-DVC code will be publicly released after review.

## 1 INTRODUCTION

Dense video captioning (DVC) has long been studied as a task that involves both temporal localization of multiple event boundaries in untrimmed videos and generating natural language descriptions for each event (Krishna et al., 2017; Wang et al., 2021; Zhou et al., 2018b). Compared to conventional video captioning (Chen et al., 2018; Xu et al., 2018; Zhang et al., 2020; Lin et al., 2022; Luo et al., 2020; Seo et al., 2022), which typically focuses on a single, short, pre-segmented clip, DVC presents unique challenges due to the necessity of accurate temporal localization of multiple events and the contextual diversity required in generating multiple event-specific captions. Recent progress in DVC has explored various strategies, including two-stage pipelines that first detect event boundaries and then generate captions, as well as methods that jointly localize and describe events (Mun et al., 2019; Krishna et al., 2017; Iashin & Rahtu, 2020; Wang et al., 2018; Yang & Yuan, 2018; Wang et al., 2020). More recently, approaches leveraging large-scale pretraining on narrated videos have demonstrated promising results by effectively capturing temporal alignments and semantic representations from extensive multimodal data (Yang et al., 2023b;a; Huang et al., 2020; Miech et al., 2019; Zellers et al., 2021; Kim et al., 2024; 2025).

Despite these advancements, existing DVC methods remain limited in their adaptability to user-specific demands. In real-world scenarios, users often require different levels of event granularity and descriptive detail when viewing videos, depending on their time constraints and objectives. For example, surveillance officers may prefer concise summaries of suspicious activities, whereas medical trainees may require fine-grained, step-by-step accounts of surgical procedures. However, as shown in Figure 1(a), the existing methods are trained on benchmarks with a fixed single annotation style, thus generating uniform outputs that overlook such diverse requirements. This lack of controllability poses a fundamental gap between current research progress and real-world usability.

To overcome this limitation, we introduce **User-Controllable Captions (UC Captions)**, a flexible dataset derived from existing DVC benchmarks (YouCook2 (Zhou et al., 2018a) and ViTT (Huang et al., 2020)) and enriched with annotations at varying levels of event granularity and descriptive

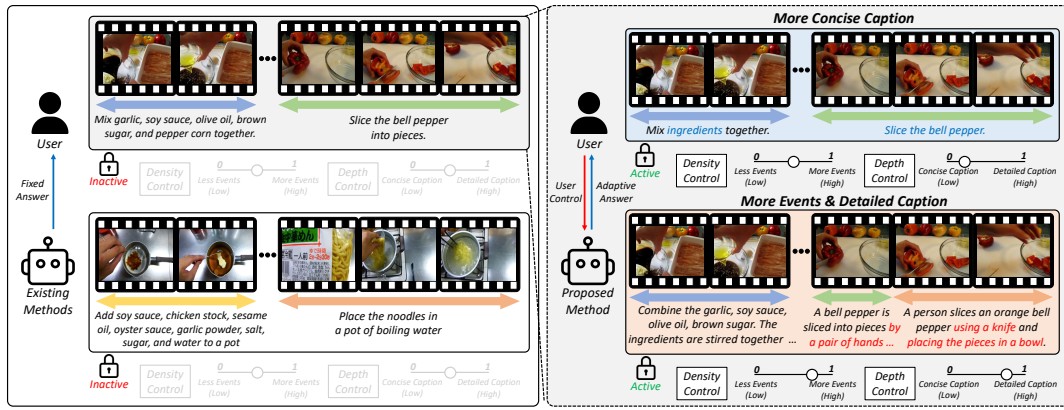

(a) *Existing Methods*                    (b) *Ours*

Figure 1: Conceptual comparison of (a) existing methods, which generates a fixed caption, and (b) ours, which enables user-controllable captions by adjusting event density and caption depth to tailor the captions to user needs.

detail. In addition, we propose **User-Controllable DVC (UC-DVC)**, a flexible baseline framework specifically designed to adapt to diverse user requirements in DVC task.

First, we construct UC Captions to provide flexible and controllable captions over video content along the following two key aspects: (*i*) **event density**, which determines how many events within a video are segmented and described, and (*ii*) **caption depth**, which specifies the level of descriptive detail for each event. To capture these aspects, we enrich existing DVC benchmarks with annotations at three levels—low, mid, and high—for both event density and caption depth. Each level is mapped to numerical control values (*i.e.,* low (0), mid (0.5), and high (1.0)), forming a structured and extensible representation of granularity. Training on these values enables the framework to learn a continuous embedding space, allowing interpolation between 0 and 1 and precise control over event density and caption depth for flexible video understanding (Figure 1(b)).

Second, we introduce UC-DVC that can adaptively generate captions based on user-specified levels of event density and caption depth. While UC Captions provides the necessary annotations, existing DVC models are limited by their architectures and cannot naturally accommodate such controllable inputs. Thus, our UC-DVC addresses this gap by incorporating user-defined values (between 0 and 1) through dedicated modules that learn density and depth representations, enabling captions that adapt to varying user demands for granularity. Consequently, it offers a flexible solution for user-controllable captions and insights for developing DVC task aligned with real-world needs.

The major contributions of our paper are as follows:

- We introduce UC Captions, the first flexible dataset that extends existing DVC benchmarks with multi-level annotations along the dimensions of event density and caption depth. This dataset enables fine-grained and continuous user control, which goes beyond the fixed annotation styles found in prior work.
- We propose UC-DVC, a baseline framework designed to incorporate user-specified density and depth values through dedicated modules. By dynamically adjusting event localization and caption generation, UC-DVC enables adaptive and user-controllable dense video captioning, offering capabilities that extend beyond those of existing models.
- Extensive experiments show that our framework establishes a new state-of-the-art on standard DVC benchmarks while also robustly outperforming baseline methods across all granularity configurations on our UC Captions benchmark. We will release the dataset and code to support future research and reproducibility.

## 2 UC CAPTIONS DATASET

### 2.1 GRANULARITY-CONTROLLABLE DATASET CONSTRUCTION

Existing DVC benchmarks (*i.e.,* YouCook2 Zhou et al. (2018a), ViTT Huang et al. (2020)) are restricted to fixed levels of granularity for a given video. As a result, they cannot adequately sup-

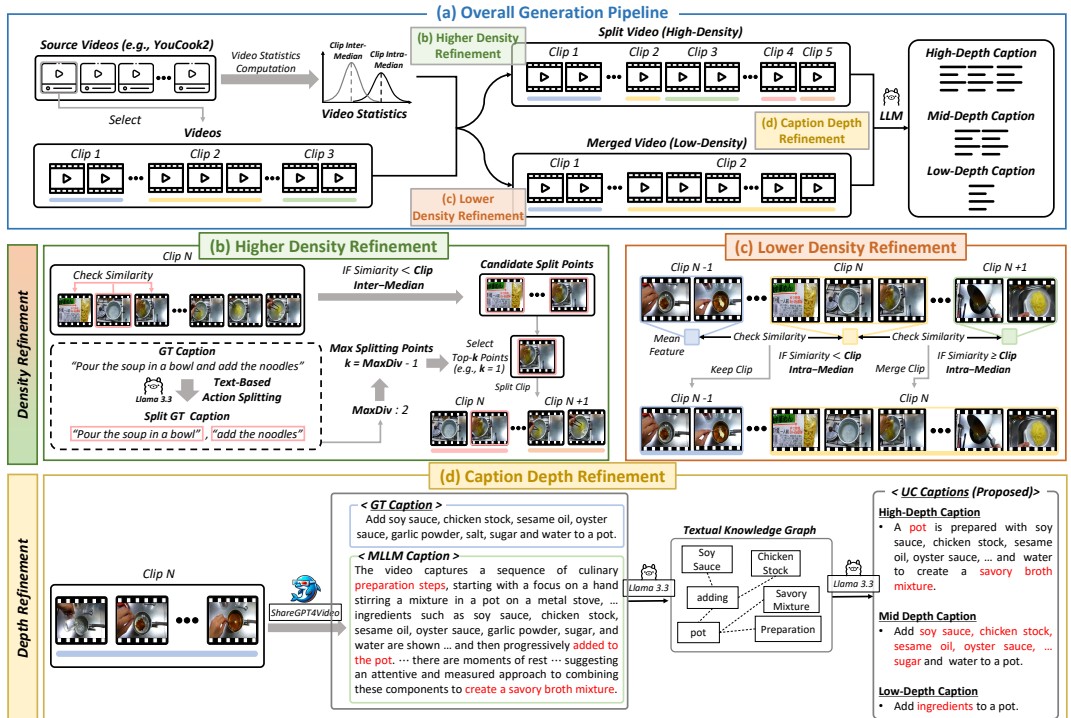

Figure 2: Generation pipeline of UC Captions. **(a) Overall Generation Pipeline** Visual similarity statistics establish data-driven thresholds for event merging (*Clip **Intra**-Median*) and splitting (*Clip **Inter**-Median*). **(b, c) Density Refinement:** Event density is controlled by merging visually similar clips or splitting incoherent ones. Splitting is further constrained by the number of actions in the ground-truth text (*MaxDiv*) to ensure semantic relevance. **(d) Caption Depth Refinement:** Caption depth is controlled by generating multi-level descriptions.

port captioning that meets diverse user requirements. To address this limitation, we extend these benchmarks and introduce the UC Captions dataset, which provides flexible and controllable annotations of video content. Specifically, we define two controllable aspects—**event density** and **caption depth**—and divide each into **three levels**, yielding a comprehensive 3×3 granularity spectrum.

We establish the varying granularity levels by analyzing the existing benchmarks. Captions in YouCook2 (Zhou et al., 2018a) consist of complete sentences that describe groups of related actions (*e.g.,* "combine soy sauce, water and sugar in a bowl"). Based on this, we define the **mid-density** level as a group of actions composing a single step, and the **mid-depth** level as a complete sentence of 7-8 words. Conversely, ViTT (Huang et al., 2020) provides concise tags for atomic actions (*e.g.,* "Adding water"). This leads to our definition of the **high-density** level as a single atomic action, and the **low-depth** level as a concise tag of 2-3 words. These two benchmarks serve as data-based anchors for our spectrum.

The remaining levels are defined by extrapolating from these anchor points. We define the **low-density** level as a major phase that covers multiple mid-density steps, representing a broader segment of activity. The **high-depth** level is defined as a detailed narrative of 18-20 words that elaborates on a mid-depth sentence, providing richer context and explanation. This completes the formal definition of our 3×3 granularity spectrum. A detailed analysis of this spectrum is provided in Table 1.

With the full spectrum defined in this data-driven manner, we construct the complete dataset through a scalable generative approach. Leveraging advanced large language models (LLMs) and multimodal large language models (MLLMs) (Chen et al., 2024a;b), our process systematically expands from the human-annotated anchors to populate the remaining levels. For instance, MLLMs generate **high-depth** captions by elaborating on mid-depth sentences or produce **low-density** captions by summarizing a sequence of mid-density events. This methodology enables the creation of a large-scale, diverse, and controllable dataset that is firmly grounded in real-world data characteristics.

Table 1: Comparison of UC Captions with existing datasets: (a) dataset statistics, (b) average event duration and word counts (avg. duration / avg. words).

| Dataset | # Captions | Annotations |
|---|---|---|
| ActivityNet (Krishna et al., 2017) | 100K | 1 Density & 1 Depth |
| YouCook2 (Zhou et al., 2018a) | 15K | 1 Density & 1 Depth |
| ViTT (Huang et al., 2020) | 56K | 1 Density & 1 Depth |
| **UC Captions (YouCook2)** | **121K** | **3 Density & 3 Depth** |
| **UC Captions (ViTT)** | **337K** | **3 Density & 3 Depth** |

(a) Statistical comparison between datasets.

| Depth Level | Density Level (YouCook2) | | | Density Level (ViTT) | | |
|---|---|---|---|---|---|---|
| | Low | Mid | High | Low | Mid | High |
| Low | 26.0 / 3.1 | 19.6 / 3.1 | 14.8 / 3.1 | 73.6 / 3.0 | 57.3 / 2.9 | _39.8 / 2.9_ |
| Mid | 26.0 / 8.4 | _19.6 / 8.6_ | 14.8 / 8.4 | 73.6 / 8.1 | 57.3 / 8.1 | 39.8 / 8.0 |
| High | 26.0 / 18.6 | 19.6 / 18.5 | 14.8 / 18.6 | 73.6 / 17.6 | 57.3 / 17.7 | 39.8 / 17.9 |

(b) Average event duration (left) and word count (right) in UC Captions (Uline: annotations from the original datasets).

## 2.2 Generation Pipeline

As shown in Figure 2, to realize the flexible granularity defined by our framework, we design a two-phase generation pipeline. This pipeline is engineered to provide precise, independent control over both event density and caption depth. Its foundation lies in a set of data-driven thresholds derived from a statistical analysis of visual similarity (*i.e.,* cosine-similarity) within our Source datasets (*i.e.,* YouCook2 or ViTT). This analysis yields two thresholds: the *Clip Intra-Median*, which captures the median similarity within a single coherent event and guides event merging, and the *Clip Inter-Median*, representing the median similarity between distinct events, which guides event splitting. This approach ensures that all granularity adjustments are contextually grounded.

**Higher/Lower Density Refinement.** The first phase leverages these pre-established thresholds to strategically adjust temporal boundaries. (*i*) To generate higher density annotations, it splits clips at points of low internal coherence where similarity drops below the *Clip Inter-Median*. To maintain semantic integrity during this splitting process, we introduce the *MaxDiv*, a constraint where an LLM parses the GT caption to identify its distinct actions (Figure 2(b)), thus aligning the number of possible splits with the number of semantic actions. Conversely, (*ii*) to generate lower density annotations, the pipeline merges adjacent clips whose visual similarity exceeds the *Clip Intra-Median*, effectively grouping them into a single, broader event (Figure 2(c)). The details about *MaxDiv* and the statistical validation for *Clip Inter-/Intra- Median* can be found in Appendix C.1.

**Caption Depth Refinement.** The second phase is responsible for generating the descriptions of varying depth (Figure 2(d)). To ensure semantic consistency and high fidelity, we introduce a **Textual Knowledge Graph (TKG)**, aligning with recent work on LLM-based knowledge graph construction (Pan et al., 2024). This strategy is motivated by findings that grounding LLM generation in structured knowledge enhances reliability and mitigates issues like hallucination (Zhang & Soh, 2024; Zhu et al., 2024). Our pipeline first uses an MLLM to generate a rich description then construct the knowledge graph using LLM. Then, we query the LLM to generate the varying level captions for each depth level with this knowledge graph. By selectively querying this structured knowledge, LLM crafts descriptions ranging from concise tags (low depth) to context-rich narratives (high depth) with high precision. It is critical for achieving consistent and fine-grained control over caption content. A detailed description of the procedure and prompting strategy is provided in Appendix C.2.

## 2.3 UC-Captions Dataset Analysis and Quality Validation

**Comparison with Existing Benchmarks.** As mentioned in Section 2.1, applying our pipeline to the YouCook2 *train*/*val* and ViTT *train*/*test* splits yields UC Captions, a large-scale resource with a 3-density by 3-depth annotation structure, as summarized in Table 1(a).

**Quantitative Validation.** Beyond its scale, we conducted a quantitative analysis to verify that UC Captions was generated as intended. The results, presented in Table 1(b), confirm the efficacy of our pipeline. As designed, increasing the density control from low to high systematically decreases the average event duration (*e.g.,* from 26.0s to 14.8s for YouCook2). Conversely, increasing the depth control from low to high consistently raises the average word count (*e.g.,* from 3.1 to 18.6). These trends validate that our pipeline provides effective and predictable control over both granularity axes, making UC Captions a reliable resource for training the generation of controllable models.

**Human Evaluation.** To additionally verify the naturalness of the generated captions, we conducted human evaluation. The detailed evaluation protocol and results are provided in Appendix D. It confirmed both the high quality of the annotations and the perceptual effectiveness of our UC Captions.

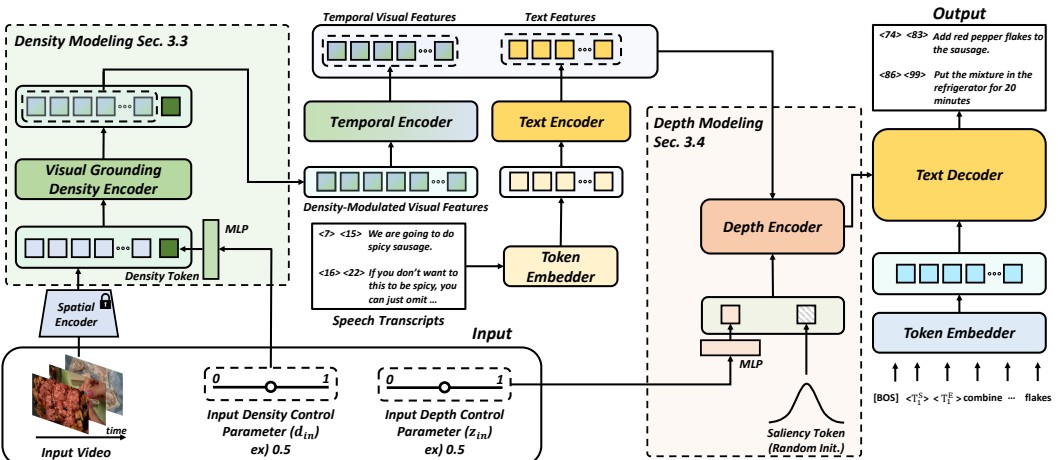

Figure 3: Network configuration of the proposed UC-DVC framework.

# 3 PROPOSED BASELINE FRAMEWORK: UC-DVC

## 3.1 PRELIMINARIES

Dense video captioning aims to temporally localize multiple events within untrimmed videos and generate corresponding natural language descriptions (Krishna et al., 2017; Wang et al., 2021). Given an input video, dense video captioning framework predicts $N$ event segments $(\langle T_s^i \rangle, \langle T_e^i \rangle)_{i=1}^{N}$, where $\langle T_s^i \rangle$ and $\langle T_e^i \rangle$ denote the start and end timestamps of the $i$-th event, and generates a caption $c_i$ for each segment.

To extract frame-level visual features from a video, we utilize a pre-trained CLIP (Radford et al., 2021) encoder that processes video frames into a sequence of visual features $V = \{v_1, v_2, ..., v_l\} \in \mathbb{R}^{l \times h}$, where $l$ represents the number of extracted frames. The visual features are then processed through a transformer-based temporal encoder, capturing both temporal dependencies between frames and contextual relationships among events. Following (Yang et al., 2023b), we also use speech text feature $S_t = \{v_1, v_2, ..., v_l\} \in \mathbb{R}^{s \times h}$, where $s$ denotes the length of speech transcripts, by encoding the speech transcript with the transformer-based text encoder. These visual and speech features serve as primary inputs for the text decoder to generate target event timestamps and caption pairs.

DVC framework jointly predicts event boundaries and captions in a sequence-to-sequence manner, generating an output sequence:

$$Z = \{(\langle T_s^n \rangle, \langle T_e^n \rangle, c_n)\}_{n=1}^{N} \tag{1}$$

where $\langle T_s^n \rangle$ and $\langle T_e^n \rangle$ represent the predicted start and end timestamps of the $n$-th event, and $c_n$ is the corresponding generated caption.

## 3.2 OVERALL ARCHITECTURE OF UC-DVC

Having established the UC Captions dataset that provides the necessary training data with multiple granularity levels, we now address the flexible baseline framework. While the dataset provides the foundation for user-controllable captioning, existing DVC frameworks are architecturally constrained and cannot inherently adapt user requirements for depth and density variation.

As shown in Figure 3, to achieve user-controllable dense video captioning, we devise UC-DVC that can incorporate density control parameter $d_{\text{in}}$ and depth control parameter $z_{\text{in}}$. Density control parameter $d_{\text{in}}$ controls the frequency of event localization, determining how many distinct events are detected and described in a given video segment. Higher density values result in finer temporal segmentation, while lower values produce coarser segmentation with fewer events. Depth control parameter $z_{\text{in}}$ modulates the level of detail in generated captions, controlling how descriptive and comprehensive each caption should be. Higher depth values produce detailed descriptions, while lower values generate concise, essential-only captions.

Both inputs are represented as floating-point values in the range [0, 1], where the three granularity levels in UC Captions correspond to: high = 1.0, mid = 0.5, and low = 0.0. These continuous values allow for fine-grained control and smooth interpolation between different granularity levels, enabling the model to dynamically adjust event localization and caption detail based on user preferences.

To capture density variation, visual features processed through the spatial encoder are passed through a visual grounding density encoder along with the density token, enabling dynamic control over event localization frequency. Subsequently, to allow adaptive modulation of caption detail level with depth variation, the visual features and speech text features are processed through a depth encoder along with the depth token. These density- and depth-modulated representations are then fed into the text decoder, which enables the model to generate variable numbers of events with user-controllable caption granularity throughout the entire pipeline. Details are described in the following subsections.

### 3.3 DENSITY MODELING

Density modeling manages how frequently events are localized in a video. The density input $d_{in} \in \mathbb{R}^1$ is first projected into a $h$-dimensional shared embedding space to generate density embedding $e_d \in \mathbb{R}^h$, represented as follows:

$$e_d = W_d d_{in} + b_d, \tag{2}$$

where $W_d \in \mathbb{R}^{h \times 1}$ and $b_d \in \mathbb{R}^h$ are learnable parameters. Then, $e_d$ is combined with the visual features $V$ and processed by a density encoder to obtain density-modulated visual features $V_d$:

$$V_d = E_d([V||e_d]). \tag{3}$$

The output features $V_d$ are then passed to the temporal encoder, resulting in temporal visual feature $\bar{V}_d$ which can dynamically modulate event localization according to the specified density level.

### 3.4 DEPTH MODELING

Moreover, depth modeling controls the level of descriptive detail in the generated captions. Similar to density modeling, the depth input $z_{in} \in \mathbb{R}^1$ is projected into an $h$-dimensional embedding space:

$$e_z = W_z z_{in} + b_z, \tag{4}$$

where $W_z \in \mathbb{R}^{h \times 1}$ and $b_z \in \mathbb{R}^h$ are learnable parameters. The depth embedding $e_z$ is combined with $\bar{V}_d$, speech text feature $S_t$ and a saliency token $s \in \mathbb{R}^h$, and then processed by a depth encoder to generate depth-modulated visual and text features $V_z$. Here, $s$ is a learnable saliency token that provides an explicit signal for depth variation, allowing the encoder to better capture and emphasize the changes in descriptive detail required at different depth levels. $V_z$ can be expressed as follows:

$$V_z = E_z([\bar{V}_d||S_t||s||e_z]). \tag{5}$$

Finally, $V_z$ is fed into the text decoder, which incorporates the density and depth specified by the input parameters $d_{in}$ and $z_{in}$ to localize diverse events in the video and generate captions with the appropriate level of descriptive detail.

### 3.5 LEARNING REPRESENTATIONS FOR DEPTH AND DENSITY

Since user requirements often vary in both event granularity and caption detail, it is crucial for the model to learn how to adapt its predictions across different depth and density levels. To enhance this adaptability, we introduce two specialized loss functions that leverage a training strategy with two ground truths from different granularity. That is, for each video sample, we construct training pairs consisting of target and reference annotations at different granularity levels. This approach enables the model to learn relative differences between density and depth variations, facilitating more robust adaptation to diverse user controls.

**Training Data Construction.** For each input video, we create training samples consisting of: (1) target sample: a density-depth pair $(d_{tar}, z_{tar})$ with ground-truth annotations $\{(\langle T_s^{i,tar} \rangle, \langle T_e^{i,tar} \rangle, c_i^{tar})\}_{i=1}^{N_{tar}}$, and (2) reference sample: a different density-depth pair $(d_{ref}, z_{ref})$ with ground-truth annotations $\{(\langle T_s^{i,ref} \rangle, \langle T_e^{i,ref} \rangle, c_i^{ref})\}_{i=1}^{N_{ref}}$, where $N_{tar}$ and $N_{ref}$ represent the number of event segments. The reference sample serves as a comparative baseline for the loss functions.

Figure 4: Explanation of proposed (a) density-aware loss and (b) depth-aware loss.

**Density-aware Loss.** We follow previous work (Yang et al., 2023b) by utilizing a loss function designed to predict current event sequence $z_{k+1} \in Z$ with given depth-modulated visual and text feature $V_z$ from the preceding sequence $(V_z, z_{1:k})$:

$$\mathcal{L}_\theta(V_z, z) = -\frac{1}{\sum_{k=1}^{L-1} w_k} \sum_{k=1}^{L-1} w_k \log p_\theta(z_{k+1} \mid V_z, z_{1:k}). \tag{6}$$

where $w_k$ indicates the weights assigned to $k$-th token. While previous methods set $w_k$ uniformly to 1, we dynamically assign $w_k$ based on IoU differences between target and reference density annotations as shown in Figure 4(a). Specifically, we set $w_k = 1/\text{IoU}_k$ where $\text{IoU}_k$ represents the IoU between target and reference timestamp of $k$-th token. We assign higher weights to timestamps with substantial density changes, enabling the model to learn how density variations affect event localization frequency by contrasting target and reference granularity levels.

**Depth-aware Loss.** The depth-aware loss ensures accurate depth-level captioning by supervising a saliency token to differentiate between target and reference caption detail levels. As shown in Figure 4(b), during training, cross-attention is applied between $V_z$ and either target or reference ground truth (GT) tokens encoded with text encoder. The saliency token distinguishes between target GT tokens (matching the intended depth level) and reference GT tokens (from different depth levels), outputting values close to 1 for target tokens and 0 for reference tokens.

To enforce this behavior, after the cross-attention operation, the token at the saliency token position is used to compute the saliency score $S$:

$$S = \sigma(W_s V_s + b_s), \tag{7}$$

where $V_s$ represents the output token at the saliency position after cross-attention, and $W_s \in \mathbb{R}^{1 \times h}$, $b_s \in \mathbb{R}$ are learnable parameters. We utilize binary cross-entropy loss to guide the saliency score as follows:

$$\mathcal{L}_s = -y \log S - (1 - y) \log(1 - S), \tag{8}$$

where $y = 1$ for target GT tokens and $y = 0$ for reference GT tokens. This comparative supervision enables the model to learn how depth variations affect caption detail by contrasting target and reference captions with different granularity levels.

## 3.6 TRAINING OBJECTIVE

The final training objective is defined as:

$$\mathcal{L} = \mathcal{L}_\theta + \lambda \mathcal{L}_s. \tag{9}$$

By jointly optimizing these two loss functions, our UC-DVC learns adaptive depth-aware caption generation and density-dependent event localization, achieving highly flexible user-controllable dense video captioning. We set $\lambda = 1$, and the performance was insensitive to variations in its value.

## 4 EXPERIMENTS

### 4.1 EXPERIMENTAL SETTINGS

**Evaluation Metrics.** Following (Yang et al., 2023b), we evaluate the performance using CIDEr (Vedantam et al., 2015) and METEOR (Banerjee & Lavie, 2005) to evaluate the captioning quality.

Table 2: Experimental results on YouCook2 *val* set (left) and ViTT *test* (right) for dense video captioning. PT denotes pre-training from the additional video datasets. † denotes results reproduced from official implementation in our environment. All methods use same CLIP as a visual encoder.

| Method | PT | UC Captions | YouCook2 (*val*) | | | | ViTT (*test*) | | | |
|---|---|---|---|---|---|---|---|---|---|---|
| | | | METEOR | CIDEr | SODA_c | F1 | METEOR | CIDEr | SODA_c | F1 |
| PDVC (Wang et al., 2021) | ✗ | ✗ | 5.56 | 29.69 | 4.94 | 26.81 | - | - | - | - |
| CM² (Kim et al., 2024) | ✗ | ✗ | 6.08 | 31.66 | 5.34 | 28.43 | - | - | - | - |
| E²DVC (Wu et al., 2025) | ✗ | ✗ | 6.11 | 34.26 | 5.39 | 28.64 | - | - | - | - |
| Streaming V2S (Zhou et al., 2024) | ✓ | ✗ | 7.10 | 32.90 | 6.00 | 24.10 | 5.80 | 25.2 | 10.00 | 35.40 |
| DIBS (Wu et al., 2024) | ✓ | ✗ | 7.51 | 44.44 | 6.39 | 31.43 | - | - | - | - |
| Vid2Seq (Yang et al., 2023b) | ✓ | ✗ | 12.30 | 67.20 | 10.30 | 32.60 | 9.50 | 50.0 | 15.00 | 46.20 |
| HiCM² (Kim et al., 2025) | ✓ | ✗ | 12.80 | 71.84 | 10.73 | 32.51 | 9.66 | 51.29 | 15.07 | 45.08 |
| Vid2Seq† (Yang et al., 2023b) | ✓ | ✗ | 12.25 | 67.51 | 10.30 | 32.17 | 9.41 | 48.85 | 14.19 | 44.93 |
| Vid2Seq† (Yang et al., 2023b) | ✓ | ✓ | 8.30(-3.95) | 41.59(-25.92) | 8.42(-1.88) | 30.96(-1.21) | 4.90(-4.51) | 10.40(-38.44) | 2.61(-11.58) | 43.15(-1.78) |
| **Vid2Seq + UC-DVC (Ours)** | ✓ | ✓ | **12.81**(+0.56) | **73.14**(+5.63) | **10.81**(+0.51) | **33.11**(+0.94) | **11.14**(+1.73) | **51.62**(+2.77) | **15.49**(+1.30) | **46.56**(+1.63) |
| HiCM²† (Kim et al., 2025) | ✓ | ✗ | 12.52 | 68.92 | 10.71 | 32.54 | 9.37 | 49.08 | 15.01 | 44.92 |
| HiCM²† (Kim et al., 2025) | ✓ | ✓ | 8.38(-4.14) | 43.72(-25.20) | 8.40(-2.31) | 31.00(-1.54) | 5.29(-4.08) | 15.27(-33.81) | 2.34(-12.67) | 42.17(-2.75) |
| **HiCM² + UC-DVC (Ours)** | ✓ | ✓ | **12.92**(+0.40) | **75.97**(+7.05) | **11.91**(+1.20) | **37.90**(+5.36) | **11.28**(+1.91) | **53.69**(+4.61) | **15.58**(+0.57) | **46.56**(+1.57) |

Table 3: Performance comparison on UC Captions on the YouCook2 and ViTT benchmarks. We adopt the most recent DVC model HiCM² and compare ours (HiCM² / HiCM² + Ours) across different density-depth levels. Note that the baseline results are obtained from **nine independent** models, each trained separately for a specific density–depth level, whereas our UC-DVC is **a single unified** model to predict all nine levels simultaneously.

| | | UC Captions (YouCook2) | | | UC Captions (ViTT) | | |
|---|---|---|---|---|---|---|---|
| Density | Metric | Low Depth | Mid Depth | High Depth | Low Depth | Mid Depth | High Depth |
| **Low** | METEOR | 9.1/**10.5**(+1.4) | 9.2/**10.3**(+1.1) | 8.4/**9.4**(+1.0) | 6.5/**7.8**(+1.3) | 8.5/**9.5**(+1.0) | 9.2/**10.1**(+0.9) |
| | F1 | 31.8/**39.0**(+7.2) | 30.2/**39.6**(+9.4) | 30.7/**39.5**(+8.8) | 44.0/**51.5**(+7.5) | 44.2/**53.8**(+9.6) | 45.1/**53.7**(+8.6) |
| **Mid** | METEOR | 10.0/**11.8**(+1.8) | 12.2/**12.9**(+0.7) | 8.8/**9.7**(+0.9) | 6.6/**8.3**(+1.7) | 9.4/**12.1**(+2.7) | 9.2/**10.0**(+0.8) |
| | F1 | 31.7/**37.6**(+5.9) | 32.1/**37.9**(+5.8) | 30.9/**36.9**(+6.0) | 45.5/**51.2**(+5.7) | 45.0/**51.1**(+5.1) | 46.1/**51.8**(+5.7) |
| **High** | METEOR | 8.4/**11.0**(+2.6) | 8.5/**10.6**(+2.1) | 7.5/**9.1**(+1.6) | 9.4/**11.3**(+1.9) | 10.8/**12.7**(+1.9) | 9.9/**11.3**(+1.4) |
| | F1 | 27.9/**30.6**(+2.7) | 27.8/**30.9**(+3.1) | 26.4/**29.9**(+3.5) | 44.9/**46.5**(+1.6) | 45.3/**48.2**(+2.9) | 45.0/**48.3**(+3.3) |

For event localization, we follow the standard protocol by computing the F1 score, the harmonic mean of average precision, and recall, across IoU thresholds of {0.3, 0.5, 0.7, 0.9}. Additionally, since these metrics do not fully reflect the overall caption coherence of dense video captioning, we also employ SODA_c (Fujita et al., 2020) to evaluate overall storytelling performance.

**Implementation Details.** Following (Yang et al., 2023b), video frames were extracted at a rate of 1 frame per second for the YouCook2 (Zhou et al., 2018a) and ViTT (Huang et al., 2020) datasets. Both the text encoder and decoder are initialized using a pre-trained T5-Base (Raffel et al., 2020) model. We utilize CLIP ViT-L/14 (Radford et al., 2021) for the visual encoder. More detailed training settings can be found in in Appendix E.

## 4.2 COMPARISON ON STANDARD BENCHMARKS

We evaluate the proposed UC-DVC with UC Captions against state-of-the-art methods (Yang et al., 2023b; Kim et al., 2025) on the original YouCook2 (Zhou et al., 2018a) and ViTT (Huang et al., 2020) datasets. As shown in Table 2, existing methods suffer performance drops when trained with UC Captions, as they lack mechanisms to model variations in density and depth. In contrast, UC-DVC effectively learns diverse expressive forms of density and depth from UC Captions, enabling robust adaptation to these controllable aspects and achieving consistent improvements on both original benchmarks. Consequently, our UC-DVC outperforms Vid2Seq and HiCM² trained on fixed annotation styles and demonstrates its practical advantage for real-world video captioning.

## 4.3 ANALYSIS WITH DENSITY AND DEPTH VARIATION

**Quantitative Analysis.** We evaluate the adaptability of UC-DVC under different density and depth inputs using the UC Captions test sets from the YouCook2 and ViTT benchmarks. The baseline HiCM² lacks mechanisms to model user-controllable aspects and thus requires **nine separate models** trained individually for different density-depth levels. As shown in Table 3, UC-DVC consistently outperforms HiCM² (Kim et al., 2025) across all configurations and metrics. The gains come from the ability of UC-DVC to leverage UC Captions to learn diverse expressive forms of density and

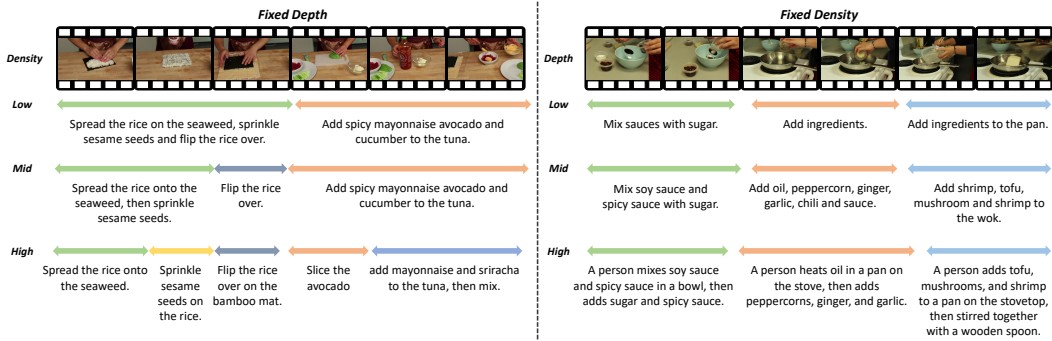

Figure 5: Visualization Results of density and depth variation of UC-DVC on YouCook2 *val* set.

Table 4: Effect of proposed component (*MaxDiv*, TKG) in data generation on YouCook2 *val* set.

| Data Type | MaxDiv | TKG | METEOR | CIDEr | SODA_c | F1 |
|-----------|--------|-----|--------|-------|--------|-----|
| **Original** | - | - | 12.80 | 71.84 | 10.73 | 32.51 |
| **UC Captions** | ✗ | ✗ | 12.82 | 72.45 | 11.25 | 34.87 |
| | ✓ | ✗ | 12.84 | 73.97 | 10.86 | 34.72 |
| | ✗ | ✓ | 12.88 | 74.39 | 11.18 | 37.25 |
| | ✓ | ✓ | **12.92** | **75.97** | **11.91** | **37.90** |

Table 5: Effect of proposed component (density and depth) on YouCook2 *val* set.

| Density | Depth | METEOR | CIDEr | SODA_c | F1 |
|---------|-------|--------|-------|--------|-----|
| ✗ | ✗ | 8.38 | 43.72 | 8.40 | 31.00 |
| ✓ | ✗ | 10.75 | 55.99 | 10.37 | 36.42 |
| ✗ | ✓ | 12.21 | 70.42 | 10.73 | 34.95 |
| ✓ | ✓ | **12.92** | **75.97** | **11.91** | **37.90** |

depth, which enables a single unified model to adapt to user-specified inputs. This approach reduces training complexity and achieves superior performance, highlighting the practical advantage of our method for real-world video captioning.

**Qualitative Analysis.** Furthermore, Figure 5 shows visualization results under varying depth and density conditions. The results clearly demonstrate that UC-DVC effectively captures and dynamically adapts to changes in both caption detail and event localization granularity, highlighting its capability to handle user requests. To provide a more comprehensive qualitative understanding of our framework, we present additional visualization results in Appendix G.

### 4.4 ABLATION STUDY

In this section, we conduct various ablation studies on YouCook2 with UC Captions using HiCM$^2$ baseline to investigate (1) effect of the visual-text knowledge graph in data generation and (2) effect of the proposed component. A detailed analysis of the generalization of the model across different backbones, the effects of our proposed losses, and computational costs is provided in Appendix F.

**Effect of the Data Generation Components.** We first investigate the effect of components in data generation. As shown in Table 4, *MaxDiv* effectively reduces excessive fragmentation, thus preventing quality degradation due to overly fine-grained segmentation. Incorporating TKG further enhances caption quality by ensuring semantic coherence and consistent depth in the captions. We achieve the highest performance when both knowledge graph methods are considered, indicating that each knowledge graph uniquely contributes to generating high-quality captions.

**Effect of the Proposed Components.** Table 5 shows the effect of the two components: Density Modeling and Depth Modeling modules. The results clearly demonstrate that both proposed components improve performance, highlighting the effectiveness of each component in uniquely contributing to event localization and captioning.

### 4.5 DISCUSSIONS

**Extensibility to MLLM Architectures.** Our architecture-agnostic control mechanism $(d, z) \in [0, 1]^2$ readily extends to decoder-only MLLMs. First, via *In-Context Learning*, control values can be verbalized as prompt tokens (e.g., `<DENSITY: 1.0>`) for direct interpretation. Second, for *Fine-Tuning*, control signals can be projected via lightweight adapters analogous to prefix tuning. This flexibility ensures our versatility of the framework amidst evolving backbone architectures.

**Complementarity with Language-Based Control.** While language-based control offers an intuitive interface, it often lacks the quantitative precision required for fine-grained generation. Our

Table 6: We report the metrics averaged across YouCook2 and ViTT. **Semantic Score** (0–1) and **Rank** (1–3, lower is better) are evaluated by Gemini-2.5-Flash on 2,000 sampled videos per each dataset.

| Dataset | Depth | Lexical Richness | Semantic Metrics | |
|---|---|---|---|---|
| | | Unique Tokens ($\uparrow$) | Avg. Score ($\uparrow$) | Avg. Rank ($\downarrow$) |
| **YouCook2** | Low | 1,058 | 0.37 | 2.98 |
| | Mid | 2,513 | 0.74 | 1.93 |
| | High | **4,730** | **0.95** | **1.10** |
| **ViTT** | Low | 6,995 | 0.26 | 2.81 |
| | Mid | 7,913 | 0.56 | 2.18 |
| | High | **14,133** | **0.87** | **1.02** |

Table 7: Breakdown of GPU hours and wall-clock time for generating full 9-level corpus.

| Dataset | # Videos | GPU Hours | Wall-clock (h) |
|---|---|---|---|
| YouCook2 | 1,686 | $\approx 69$ | $\approx 34.5$ |
| ViTT | 7,461 | $\approx 306$ | $\approx 153.0$ |
| **Total** | **9,147** | $\approx 375$ | $\approx 187.5$ |

Table 8: Comparisons of training time, inference time, and number of parameters.

| Method | Train (s) (*per iter*) | Infer (s) (*per video*) | # Params |
|---|---|---|---|
| Vid2Seq (Yang et al., 2023b) | 0.208 | 0.681 | 289.2M |
| **Vid2Seq+Ours** | 0.229 | 0.705 | 324.7M |
| HiCM$^2$ (Kim et al., 2025) | 0.941 | 0.904 | 290.3M |
| **HiCM$^2$+Ours** | 0.986 | 0.933 | 325.8M |

continuous control mechanism is not intended to replace language queries but to serve as a complementary tool that provides *precision* (enabling smooth interpolation) and *unambiguity* (avoiding interpretational variability). Ideally, these paradigms can be unified; future work could learn to map natural language instructions (*e.g.,* "Briefly summarize.") to coordinates in our $(d, z)$ space, combining the user-friendliness of language with the consistency of our control framework.

**More Analysis in UC-Captions.** Table 6 demonstrates that our method produces meaningfully richer content, not merely longer text. First, regarding **lexical richness**, caption depth correlates strongly with the number of Unique Tokens. For instance, the High depth level in YouCook2 shows a 4.5-fold increase (1,058 to 4,730) over the Low level, and the vocabulary size in ViTT doubles (6,995 to 14,133). This confirms that deeper captions utilize a systematically richer vocabulary. Second, regarding **semantic richness**, LLM evaluations on 2,000 videos reveal a robust alignment between depth and semantic value. Semantic scores increase monotonically (*e.g.,* YouCook2: 0.37 to 0.95), which is consistently reflected in the ranking analysis, improving from $\sim$2.9 (Low) to near-perfect scores of $\sim$1.0 (High). These results validate that our framework effectively modulates both lexical diversity and meaningful information content.

**Computational Costs.** We report the computational costs for our framework. First, regarding **dataset generation**, Table 7 presents the computational cost for creating our dataset. Generating the full corpus of 9,147 videos required approximately 375 GPU-hours. Using 2 RTX A6000 GPUs, the total wall-clock time was approximately 187.5 hours, involving three density passes per video to produce all 9 density-depth combinations. Second, regarding **model training and inference**, we compare the training and inference times of our method with Vid2Seq (Yang et al., 2023b) and HiCM$^2$ (Kim et al., 2025) in Table 8. Incorporating two additional user inputs—depth and density—our method slightly increases training and inference times. However, considering the significant advantage of dynamically handling user-controllable depth and density inputs, we believe this additional computational cost is marginal and highly justified for practical scenarios.

## 5 CONCLUSION

In this paper, we proposed a novel framework for user-controllable dense video captioning that dynamically adapts to varying levels of event localization density and caption depth. To this end, we presented UC Captions, a large-scale dataset that enables flexible control over these two critical dimensions: density and depth. Our UC-DVC network successfully understands and utilizes depth and density variations according to user demands through density and depth modeling modules and two specialized loss functions. We believe that our dataset and method not only enhance dense video captioning but also potentially serve as a useful resource for future work exploring other video understanding tasks.

## ETHICS STATEMENT

Our work introduces UC-DVC, which enables explicit control over event density and caption depth. We believe this has clear positive impacts: (*i*) it promotes transparent, user-steerable video understanding that can better match accessibility needs (*e.g.,* coarse summaries for quick scanning, detailed descriptions for accessibility) (*ii*) it provides a controlled research testbed for studying over/under-segmentation, hallucination, and bias in vision–language systems and (*iii*) it encourages reproducible evaluation across a full spectrum of granularities rather than single "one-size" settings. In line with the ICLR Code of Ethics, we aim for responsible stewardship, scientific rigor, and openness in reporting.

**Risks, misuse, and our mitigation.** As with any video understanding technology, UC-DVC could be misused to generate persuasive but misleading narratives or to aid surveillance-style analytics. To reduce these risks, we (a) release only annotations and code, never original media (b) license our "UC Captions" strictly for research, prohibiting commercial use, re-identification, and any attempt to link annotations to individuals.

**Data Provenance and Licensing.** We build on public datasets (*e.g.,* YouCook2, ViTT) strictly under their terms. We do not redistribute videos our release contains JSON annotations referencing public identifiers only. All synthetic captions are marked as derivative research artifacts and distributed under a research-only license requiring attribution and forbidding re-identification.

**Privacy.** We do not store Personally Identifiable Information (PII).

**Human subjects and integrity.** Human evaluations are conducted with informed consent, clear task instructions, fair compensation, and minimal data collection. No sponsor influenced study design, analysis, or publication decisions. Where required, we followed institutional ethics/IRB processes and comply with determinations.

**Legal compliance and transparency.** We comply with applicable copyright and data-protection laws and adopt stricter-than-minimum privacy practices consistent with the ICLR Code of Ethics. Our code, prompts, and hyperparameters will be released to enable reproducibility.

## REPRODUCIBILITY STATEMENT

To ensure the reproducibility of our work, we provide detailed descriptions of our data generation pipeline and experimental setup. The comprehensive methodology for constructing the UC Captions dataset, including the data-driven thresholds for density refinement and the Textual Knowledge Graph (TKG) approach for depth refinement, is fully detailed in Appendix C. All implementation details for our UC-DVC framework, such as model architecture, training hyperparameters, and the computational environment, are specified in Appendix E. Upon acceptance, we will publicly release the complete source code, our model weights, and the full UC Captions dataset to facilitate verification and further research in the community.

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

# Appendix

## A THE USE OF LARGE LANGUAGE MODELS (LLMS)

We utilized Large Language Models (LLMs) during the preparation of this manuscript. The application of these models was strictly confined to improving the quality of the written text, such as correcting grammatical errors and proofreading for typos. The core contributions of this paper, including its foundational ideas, experimental design, analysis, and conclusions, are entirely the original work of the human authors.

## B RELATED WORK

### B.1 DENSE VIDEO CAPTIONING

Dense Video Captioning involves detecting events in untrimmed videos and generating captions for them (Krishna et al., 2017; Wang et al., 2021; Zhou et al., 2018b). Traditional methods first localize events and then generate captions separately (Mun et al., 2019; Krishna et al., 2017; Iashin & Rahtu, 2020; Wang et al., 2018; Yang & Yuan, 2018; Wang et al., 2020). This pipeline does not account for the interaction between the two tasks. To address this, recent studies explore end-to-end models that learn both tasks together. PDVC (Wang et al., 2021) formulates captioning as a set prediction task using DETR (Zhu et al., 2020) to predict event intervals and captions simultaneously. Vid2Seq (Yang et al., 2023b) treats it as a sequence-to-sequence problem, generating timestamps and captions in a single output while incorporating speech-to-text. Other works further refine the process. One approach introduces visual memory and streaming decoding (Zhou et al., 2024), while others address specific biases. For instance, $E^2$DVC (Wu et al., 2025) tackles the problem of temporal bias, where models overlook short-duration events, by using a non-trainable clustering module to ensure all potential events are treated equally. Another line of research enhances captioning through retrieval-augmented generation. $CM^2$ (Kim et al., 2024) first introduced this idea, and $HiCM^2$ (Kim et al., 2025) further develops it by constructing a hierarchical compact memory inspired by human cognition to retrieve more relevant knowledge. Furthermore, recent advances in multimodal large language models improve video understanding. TimeChat (Ren et al., 2024) introduces a timestamp-aware encoder and a sliding Q-Former, while VTimeLLM (Huang et al., 2024) applies boundary-aware training to refine event localization. Despite these improvements, existing methods assume a fixed level of detail in captions, limiting flexibility. To address this, we introduce a synthetic dataset that reflects variable caption density and propose a novel framework that dynamically adjusts captions based on user specifications.

### B.2 SYNTHETIC DATA GENERATION

Recent synthetic data generation via LLM and MLLM to produce high-quality training data has been receiving significant attention for its potential to enhance model performance and reduce the need for manually annotated datasets (Chen et al., 2024a; Zhang et al., 2024; Ding et al.; Wang et al., 2022; Xu et al., 2023). Notably, ShareGPT4V (Chen et al., 2024a) introduces a large-scale dataset of 1.2M detailed image captions generated using GPT-4V, significantly enhancing multimodal alignment in large vision-language models. Similarly, ShareGPT4Video (Chen et al., 2024b) employs a differential sliding-window captioning strategy to generate temporally precise and context-rich video captions, improving both video understanding and generation tasks. These approaches have demonstrated substantial performance gains across multiple benchmarks, highlighting the effectiveness of synthetic data in multimodal learning. Given that DVC requires annotation of untrimmed video sequences, manual labeling is particularly expensive and time-consuming. To this end, we introduce UC Captions specifically designed for controllable density and depth variations, leveraging MLLMs and LLMs to enable more flexible and adaptable dense video captioning. Our dataset facilitates systematic evaluation across multiple granularity levels, addressing a key limitation of existing DVC benchmarks and fostering progress toward user-controllable video understanding.

Table 9: Key percentiles of intra-group (for merging) and inter-group (for splitting) similarity distributions. The clear separation validates the use of their medians as robust thresholds.

| Dataset | Intra-Group | | Inter-Group | |
|---|---|---|---|---|
| | 10-th Perc. | Median | Median | 90-th Perc. |
| YouCook2 | 0.8945 | 0.9459 | 0.7950 | 0.8581 |
| ViTT | 0.8647 | 0.9297 | 0.8312 | 0.9111 |

Table 10: ANOVA test results for the average event duration across the three density levels. The extremely low P-values indicate that the means of the groups are statistically distinct.

| Dataset | F-statistic | P-value |
|---|---|---|
| UC Captions (YouCook2) | 817 | $p < 0.001$ |
| UC Captions (ViTT) | 6905 | $p < 0.001$ |

## C ADDITIONAL PIPELINE AND VALIDATION DETAILS FOR UC CAPTIONS

This section provides detailed implementation specifics and statistical validation for the core mechanisms of our data generation pipeline, organized thematically.

### C.1 DENSITY REFINEMENT: IMPLEMENTATION AND VALIDATION

This subsection details the mechanisms used to control event density, including the data-driven thresholds and text-aware constraints, along with their statistical validation.

**Visual Similarity Thresholds.** As mentioned in Section 2.2, the foundation of our pipeline lies in data-driven visual similarity thresholds. These are established by analyzing the cosine similarity of clip features from our anchor datasets. "Intra-Group" pairs consist of adjacent clips from the same ground-truth event, while "Inter-Group" pairs are from different events. The median similarity of each group is calculated to establish the *Clip Intra-Median* (for merging) and *Clip Inter-Median* (for splitting).

As detailed in Table 9, the analysis shows a clear separation between these two distributions. For both datasets, the median similarity of Inter-Group clips is lower than the 10-th percentile of Intra-Group clips. Conversely, the median similarity of Intra-Group clips is higher than the 90-th percentile of Inter-Group clips. This demonstrates that the median of each distribution lies outside the primary range of the other, which validates our use of their respective medians as highly reliable, data-driven thresholds.

**MaxDiv.** To maintain semantic integrity during the splitting process, we introduce *MaxDiv*, a text-aware constraint, as mentioned in Section 2.2. While low visual similarity suggests a potential boundary for a new event, relying on visual cues alone can produce meaningless segments. The *MaxDiv* constraint prevents this by ensuring that any temporal split is grounded in the semantic content of the source caption.

This process is directly illustrated in Figure 2(b). We first perform **Text-Based Action Splitting**, where a powerful LLM (Llama 3.3-70B) parses the original GT Caption into its constituent action phrases. For example, the caption "*Pour the soup in a bowl and add the noodles*" is decomposed into two distinct actions: "*Pour the soup in a bowl*" and "*add the noodles*". The number of these resulting phrases sets the value of *MaxDiv* (in this case, *MaxDiv* = 2). From this, we determine the maximum number of visual splits to perform, $k = MaxDiv - 1 = 1$. The pipeline then analyzes the video clip to identify candidate split points based on visual dissimilarity, and selects up to the top-$k$ points to perform the split. It is important to note that *MaxDiv* serves as an upper bound, not a strict target. If the text describes multiple actions that occur within a visually seamless sequence with few candidate split points, the pipeline will perform fewer than $k$ splits, prioritizing the visual evidence. This method ensures that the final visual segments directly correspond to the semantic actions identified by the LLM.

**Validation of Generated Density Levels.** To validate that our density refinement process creates statistically distinct groups, we performed a one-way ANOVA test. For this test, we formulated a null hypothesis ($H_0$) which posits that there is no meaningful difference in the mean event durations across the Low, Mid, and High density levels. As summarized in Table 10, the extremely low p-values (p<0.001) lead to a strong rejection of this null hypothesis. This result confirms that our density control mechanism is effective, producing event segmentations that are demonstrably and significantly different from one another.

## C.2    CAPTION DEPTH REFINEMENT

This subsection details our novel approach for generating multi-level captions using a Textual Knowledge Graph (TKG).

**Textual Knowledge Graph (TKG) for Caption Depth Refinement.** To ensure semantic consistency and high fidelity across all generated caption levels, the second phase of our pipeline (Section 2.2), Caption Depth Refinement, centers on a **TKG**. This approach is motivated by recent advancements showing that grounding LLM generation in structured knowledge enhances reliability and reduces issues like hallucination (Pan et al., 2024; Zhang & Soh, 2024; Zhu et al., 2024). Our process utilizes a combination of state-of-the-art models. We first employ **ShareGPT4Video** (Chen et al., 2024b) as the primary MLLM to generate an initial, rich synthetic caption, followed by **Llama 3.3-70B** (Grattafiori et al., 2024) as the primary LLM to refine this output into three distinct levels of detail. All data generation processes were conducted using one NVIDIA RTX A6000 GPU for each model.

The refinement process begins when the LLM builds the TKG using the original **GT Caption** as an anchor. The LLM first analyzes the GT Caption to extract its core actions and entities. It then uses information from the more detailed **MLLM Caption** to add supplementary details. The LLM constructs the final TKG by adding the details from the MLLM caption to the core facts from the GT, following the instruction to "**Build a Knowledge Graph (KG)**". This approach is used to control for potential MLLM hallucinations while still using its descriptive information.

For example, given the GT Caption "*fry some minced garlic roasted red chili...*", the LLM identifies 'frying' as a core action. It then adds details from the MLLM Caption, such as that the garlic is fried "*until golden*" and the sauce is "*thickening*". This results in a KG containing detailed and grounded facts, such as '("minced garlic", "frying", "golden")'.

Once this event-specific knowledge graph is constructed, the final caption depth refinement becomes a structured querying task. For instance, to generate a **High Depth** caption, the LLM is prompted to use more details from the graph, while a **Low Depth** caption uses only the most essential actions and entities. This TKG-based method provides several critical advantages by ensuring semantic consistency since all levels are derived from the same facts, offering precise control by turning generation into a structured query, and significantly enhancing reliability by grounding the output of the LLM to reduce the risk of hallucination. Table 11 presents the detailed prompt, including the exact instructions and few-shot examples given to the LLM, which ensures the clarity and reproducibility of our approach.

Table 11: The comprehensive prompt used to guide the LLM in generating captions at three distinct depth levels. It includes the overall task, specific instructions for each level based on querying a knowledge graph, and few-shot examples.

---

**You are an advanced AI model specialized in Video Captioning. Your task is to generate three different levels of captions from a given Ground Truth (GT) caption and Multimodal-Large-Language Model (MLLM) caption.**

*Task Instructions*:

1. **Analyze the GT caption** to extract the core actions and objects.

2. **Extract key information** from the MLLM caption to enrich the GT caption.

3. **Based on the GT Caption, build a Knowledge Graph (KG)** to represent the relationships between entities and actions.

4. **Generate captions at three different levels by querying the KG:**

- **Low Depth**: Generate a 2-3 word tag representing the core action and object.

- **Mid Depth**:

– Case 1: If "(Merged)" or "(Part)" is in the GT caption, generate a 7-8 word sentence describing the primary interaction.

– Case 2: Otherwise, return the original GT caption.

- **High Depth**: Generate a 18-20 word sentence incorporating all relevant entities and relationships.

*Few-shot Examples*:

**Example 1 (Merged) or (Part) in GT caption**:

**GT Caption:**
"fry some minced garlic roasted red chili and the sauce and add in the corn starch and water mixture (merged)"

**MLLM Caption:**
"The video captures a culinary scene where garlic is fried until golden, and then red chili peppers are added to heat through. Corn starch and water are stirred into a pot of sauce, thickening it as it cooks."

**KG (Knowledge Graph)**

```
[
    {"minced garlic", "frying", "golden"},
    {"roasted red chili", "frying", "heat through"},
    {"sauce", "corn starch", "water", "stirring",
     "pot", "thickening", "cooking"}
]
```

**Output:**

**Low Depth**
"Fry ingredients."

**Mid Depth**
"Fry minced garlic and red chili, then add sauce."

**High Depth**
"Minced garlic is fried until golden before adding roasted red chili peppers, sauce mixture is then combined with cornstarch and water."

**Example 2 (No Merged or Part in GT caption)**:

...

---

Table 12: Human evaluation results. (a) shows the annotation quality, measured by Temporal Coherence (TC %) (left) and Caption Quality (CQ Score (1-3)) (right). (b) presents the control mechanism effectiveness, showing density control accuracy and depth quality scores. Underlined fonts indicate the anchor of each dataset.

| Dataset | Depth Level | Density Level | | |
|---|---|---|---|---|
| | | Low | Mid | High |
| YouCook2 | Low | 96.4 / 2.52 | 92.9 / 2.93 | 96.4 / 2.82 |
| | Mid | 96.4 / 2.93 | 100 / 2.93 | 89.3 / 2.56 |
| | High | 89.3 / 2.93 | 100 / 2.75 | 96.4 / 2.64 |
| ViTT | Low | 92.9 / 2.54 | 82.1 / 2.61 | 89.3 / 2.45 |
| | Mid | 96.4 / 2.89 | 92.9 / 2.54 | 80.6 / 2.71 |
| | High | 85.7 / 2.93 | 96.4 / 2.64 | 100 / 2.75 |

(a) Annotation Quality.

| Dataset | Control Metric | Level | Score |
|---|---|---|---|
| YouCook2 | Density Control (%) | $\underline{\text{Mid}} \to$ Low | 96.4 |
| | | $\underline{\text{Mid}} \to$ High | 88.1 |
| | Depth Quality (1-3) | Low | 2.86 |
| | | Mid | 2.71 |
| | | High | 2.88 |
| ViTT | Density Control (%) | $\underline{\text{High}} \to$ Mid | 89.3 |
| | | $\underline{\text{High}} \to$ Low | 97.6 |
| | Depth Quality (1-3) | Low | 2.79 |
| | | Mid | 2.86 |
| | | High | 2.98 |

(b) Control Mechanism Effectiveness.

# D   HUMAN EVALUATION FOR UC CAPTIONS DATASET

**Human Evaluation Protocol.** Annotators were presented with video clips and their corresponding captions. Depending on whether the clip's density or depth level differed from the dataset's anchor configuration (mid-density, mid-depth for YouCook2; high-density, low-depth for ViTT), they were asked a series of questions designed to evaluate different aspects of our data generation pipeline:

- **Temporal Coherence (Q1):** Annotators rated whether the video segment's flow was natural. This assesses the quality of our automated segmentation for varying density levels. A "Yes" indicates a perceptually coherent clip.
- **Caption Quality (Q2):** Annotators judged the factual correctness and relevance of the caption on a 3-point scale (1: Inappropriate, 2: Ambiguous, 3: Appropriate). This validates the quality of our caption depth refinement.
- **Control Mechanism Validation (Q3 & Q4):** In cases of density or depth variation, annotators performed a comparative judgment against an anchor video or caption. They were asked if the current sample correctly reflected the intended change (*e.g.,* "Does the current video contain more actions?", "Is the current caption more detailed?"). This directly measures the perceptual success of our control dimensions.
- **Absolute Depth Assessment (Q5):** Finally, annotators determined if a caption's detail level was appropriate for its given depth level (Low, Mid, High), with options to classify it as needing to be more concise, appropriate, or more detailed.

To validate our dataset, we conducted a human evaluation to assess two key aspects: (1) the quality of the generated annotations and (2) the perceptual effectiveness of our control mechanisms. We recruited 14 experts from 4 different institutions to ensure a diverse evaluation pool. Each expert evaluated 36 randomly sampled video-caption pairs, resulting in a total of 504 evaluations.

**Annotation Quality.** We first evaluated the quality of the generated annotations, focusing on the temporal coherence of video segments and the appropriateness of captions. As shown in Table 12(a), the generated video clips maintain a high degree of temporal coherence. For YouCook2, scores were consistently high, averaging over 95%, while for ViTT, they also demonstrated strong coherence, averaging over 90% across all levels. Furthermore, the caption quality scores for all generated levels are comparable to the original human-annotated anchors (*e.g.,* 2.93 for the YouCook2 anchor vs. 2.52-2.93 for others). These results indicate that our pipeline generates high-quality, reliable annotations across the entire 3×3 granularity spectrum.

**Control Mechanism Effectiveness.** We then assessed the perceptual effectiveness of our density and depth control mechanisms. The results in Table 12(b) show that our density control is perceptually effective, achieving high accuracy scores for both YouCook2 and ViTT (e.g., 97.6% for ViTT's High→Low control). The depth control was also validated, with captions at all levels receiving high appropriateness scores. This confirms that our framework successfully generates captions tailored to the specified level of detail, validating that our dataset is effectively designed to support the training of perceptually meaningful, controllable systems.

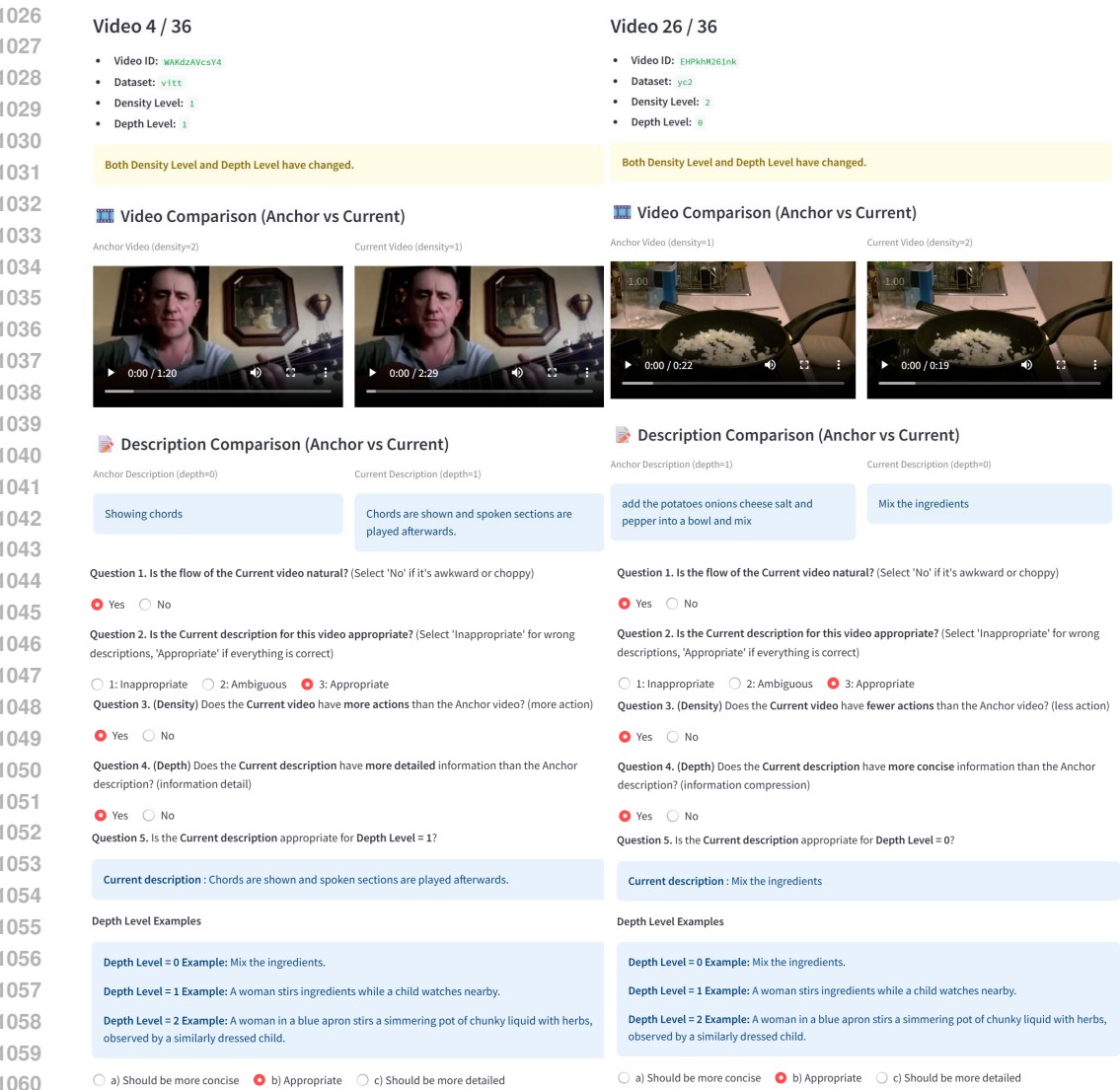

Figure 6: Visualization of our human evaluation protocol.

## E  ADDITIONAL IMPLEMENTATION DETAILS FOR UC-DVC

As described in the main paper, we utilized the pretrained T5-Base model with embedding size $h = 768$ for the text encoder and decoder, while the visual encoder was based on the pretrained CLIP ViT-L/14 model. All experiments were implemented using PyTorch, with a batch size of 2 and an initial learning rate of 3e-4. For each density and depth encoder ($E_d$ and $E_z$), we use 3 encoder transformer layers with same hidden dimension $h = 768$ of T5-Base model. We employed the Adam optimizer (Kingma, 2014), setting parameters to (0.9, 0.999), consistent with standard practices. During training, video frames were sampled at a consistent rate of 1 frame per second (fps) and standardized into sequences of 100 frames through padding or subsampling. Training was conducted on a single NVIDIA RTX 4090 GPU to 20 epochs for UC Captions of YouCook2 and ViTT datasets.

## F  ADDITIONAL EXPERIMENTS

**Evaluating Generalization Across Different MMLMs and LLMs.** To validate the generalization capability of our proposed UC-DVC framework, we conducted additional experiments

Table 13: Experimental results on the YouCook2 *val* set using different Large Language Models (LLMs).

| Method | MLLM | LLM | METEOR | CIDEr | SODA_c | F1 |
|---|---|---|---|---|---|---|
| HiCM[2] | - | - | 12.80 | 71.84 | 10.73 | 32.51 |
| Ours | InternVideo2 | Llama 3-8B | 12.62 | 72.58 | 10.92 | 35.87 |
| | InternVideo2 | Llama 3.3-70B | 12.75 | 74.15 | 11.28 | 36.94 |
| | ShareGPT4Video | Llama 3-8B | 12.79 | 74.82 | 11.45 | 37.18 |
| | ShareGPT4Video | Llama 3.3-70B | **12.92** | **75.97** | **11.91** | **37.90** |

Table 14: Effect of proposed losses $\mathcal{L}_\theta$ and $\mathcal{L}_s$ on YouCook2 *val* set.

| $\mathcal{L}_\theta$ | $\mathcal{L}_s$ | METEOR | CIDEr | SODA_c | F1 |
|---|---|---|---|---|---|
| ✗ | ✗ | 10.47 | 67.58 | 10.03 | 35.32 |
| ✓ | ✗ | 11.10 | 71.66 | 10.81 | 37.66 |
| ✗ | ✓ | 12.31 | 73.68 | 11.31 | 36.22 |
| ✓ | ✓ | **12.92** | **75.97** | **11.91** | **37.90** |

using different MLLM and LLM backbones. Specifically, we experimented with InternVideo2 (Wang et al., 2024) and ShareGPT4Video (Chen et al., 2024b) for MLLM and Llama 3-8B and Llama 3.3-70B for LLM. As shown in Table 13, our method achieves robust performance across all MLLM and LLM combinations, indicating that the effectiveness of our method is not reliant on a specific MLLM and LLM. Notably, UC-DVC consistently achieves competitive results regardless of the MLLM and LLM choice, demonstrating its strong generalizability and adaptability.

**Effect of the Proposed Losses.** We further analyze the effect of the proposed density-aware and depth-aware losses. To assess their contributions, we performed an ablation study on the YouCook2 dataset. Results shown in Table 14 indicate that each loss individually enhances the performance in event localization and captioning metrics. When evaluating without the density-aware loss, we set the weights $w_k$ uniformly to 1, following the conventional setting of prior works (Kim et al., 2025). Specifically, the density-aware loss notably improves the accuracy and granularity of event boundary predictions, while the depth-aware loss substantially enhances caption quality by ensuring more detailed and contextually coherent descriptions. The combined use of both losses achieves the best overall performance, confirming the effectiveness of our proposed loss functions in guiding the model to better understand and utilize density and depth variations.

# G  ADDITIONAL VISUALIZATION RESULTS

**Visualization Results under Varying Depth and Density on the ViTT Dataset.** Figure 7 visualizes the performance of UC-DVC on the ViTT *test* set, which contains general instructional videos such as musical instrument tutorials and makeup application. The left panel shows that as event density increases, the model successfully identifies more fine-grained events (*e.g.,* from one general "lesson" segment to four distinct actions for playing guitar). The right panel shows that as caption depth increases, the descriptions become progressively more detailed (*e.g.,* from "Eye gets primed" to a full sentence describing the application of eyeshadow with a specific tool). These results are significant as they demonstrate the robustness and generalizability of our framework beyond the cooking domain, proving its effectiveness on a wider variety of real-world video content.

**Comparison with State-of-The-Art Framework.** Figure 8 provides qualitative comparisons among ground-truth annotations, predictions from HiCM[2], and our proposed UC-DVC. While both models predict meaningful event boundaries and captions, UC-DVC consistently produces more accurate and contextually rich captions and boundaries compared to HiCM[2]. This demonstrates that our proposed loss functions effectively allow the model to leverage additional semantic information from various density and depth annotations during training.

**Fine-grained Control via Interpolation.** Figure 9 and 10 demonstrate our model's ability to handle fine-grained, continuous control via interpolation, going beyond the discrete Low(0.0), Mid(0.5), and High(1.0) levels. Figure 9 shows how the model generates an intermediate number of events for a density value of 0.25—more detailed than Low but less segmented than Mid. Similarly, Figure 10 illustrates how the model produces captions for a depth value of 0.75 that are more descriptive than Mid-level sentences but not as verbose as High-level narratives. These visualizations confirm that UC-DVC has learned a smooth, continuous control space, allowing for precise adjustments that align with nuanced user preferences.

**Visualization of Visual-Text Knowledge Graph.** In Figure 11, we visualize the generated visual-text knowledge graphs used in our UC Captions dataset generation pipeline. These graphs clearly illustrate how visual and textual information are aligned, effectively guiding the generation

of coherent and contextually accurate captions. Such visualizations demonstrate the capability of our method to integrate multimodal information, thereby facilitating fine-grained control over captioning granularity.

**Visualization of General Domain Cases.** Figure 12 presents the qualitative results of UC-DVC on general domain videos, extending beyond the culinary focus of YouCook2. The figure demonstrates the capability of the model to accurately localize and describe diverse activities, ranging from intricate procedural steps in fishing (e.g., "Adding a knot", "Dropping the line in") to precise body movements in physical exercises (e.g., "Standing on toes", "Rotating the knee out"). These results highlight the robustness and generalizability of our framework, confirming that our dense captioning capability effectively transfers to a wide range of real-world scenarios involving complex temporal events.

**Visualization of Streaming Scenario.** Figure 13 validates the adaptability of UC-DVC in a real-time streaming scenario where control requirements change dynamically. In this visualization, we shift the density input from 0.5 (Mid) to 1.0 (High) during the continuous video stream. The model demonstrates immediate responsiveness, seamlessly transitioning from generating broader event descriptions to capturing fine-grained actions as the density signal increases. This confirms that our framework allows for modulation of event granularity, making it highly effective for interactive applications where user needs may fluctuate continuously.

**Visualization of Failure Cases.** Figure 14 analyzes representative error modes encountered by our framework. The left panel illustrates an instance of *over-segmentation* combined with *temporal state hallucination*. While the ground truth defines a single continuous event, the model incorrectly fragments it into two segments. Furthermore, the visual prominence of "boiling water" in the latter frames retroactively influences the first segment, causing the model to hallucinate that the initial water is already "boiling" before the heating process completes. The right panel depicts an *under-segmentation* error where two temporally distinct atomic actions (adding oregano and adding basil) are merged into a single coarse caption due to their high visual similarity. These cases highlight the challenges in maintaining physical state consistency across segmented events and distinguishing visually analogous actions in rapid succession.

**Impact of Control Parameters on Latent Representations.** To further validate that our control parameters effectively influence the learned representations, we visualize the latent representations produced by the depth encoder using t-SNE dimensionality reduction in Figure 15. We extract features corresponding to depth values of 0.0, 0.5, and 1.0 from the depth encoder and project them into a two-dimensional space with t-SNE. The visualization reveals clear clustering patterns: features from the same depth level are tightly grouped together, while features from different depth levels are distinctly separated in the latent space. This demonstrates that our depth encoder successfully learns depth-dependent representations, confirming that the control parameters have a substantial and meaningful impact on the latent representations. Such separation in the feature space enables the model to generate captions with varying levels of descriptive detail according to user demand.

**Visualization of Difficult Scenarios in UC Captions.** Figure 16 presents representative examples from our UC-Captions, contrasting two difficult scenarios: *Low Density & Low Depth* and *High Density & High Depth*. As shown in the visualization, the dataset successfully generates distinct annotations tailored to each specific requirement. In the *Low Density & Low Depth* case, the captions provide concise summaries of broader event segments. Conversely, in the *High Density & High Depth* setting, the annotations accurately localize fine-grained atomic actions while offering rich, detailed narratives. Crucially, this comparison demonstrates that our data generation pipeline maintains high quality and factual consistency across these contrasting settings, producing hallucination-free captions regardless of the granularity complexity.

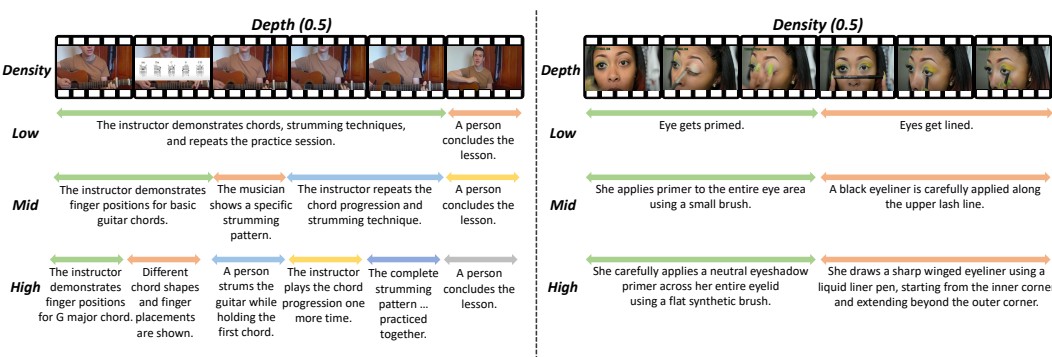

Figure 7: Visualization Results of density and depth variation of UC-DVC on ViTT *test* set.

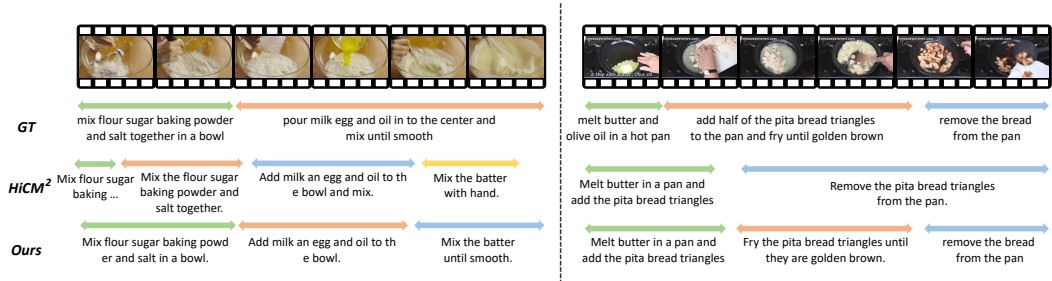

Figure 8: Visualization Results for YouCook2 *val* set. We compare our method with HiCM$^2$.

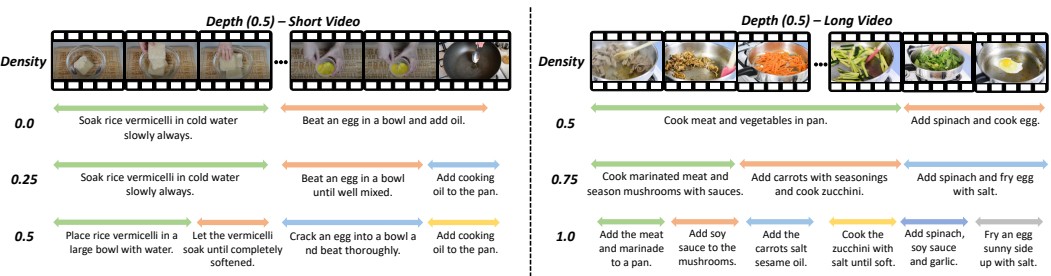

Figure 9: Visualization Results for intermediate density levels on the YouCook2 *val* set. The figure shows outputs for density values of 0.25 and 0.75.

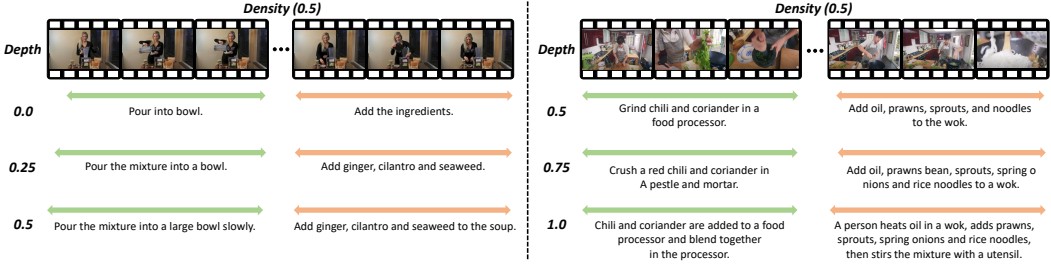

Figure 10: Visualization Results for intermediate density levels on the YouCook2 *val* set. The figure shows outputs for depth values of 0.25 and 0.75.

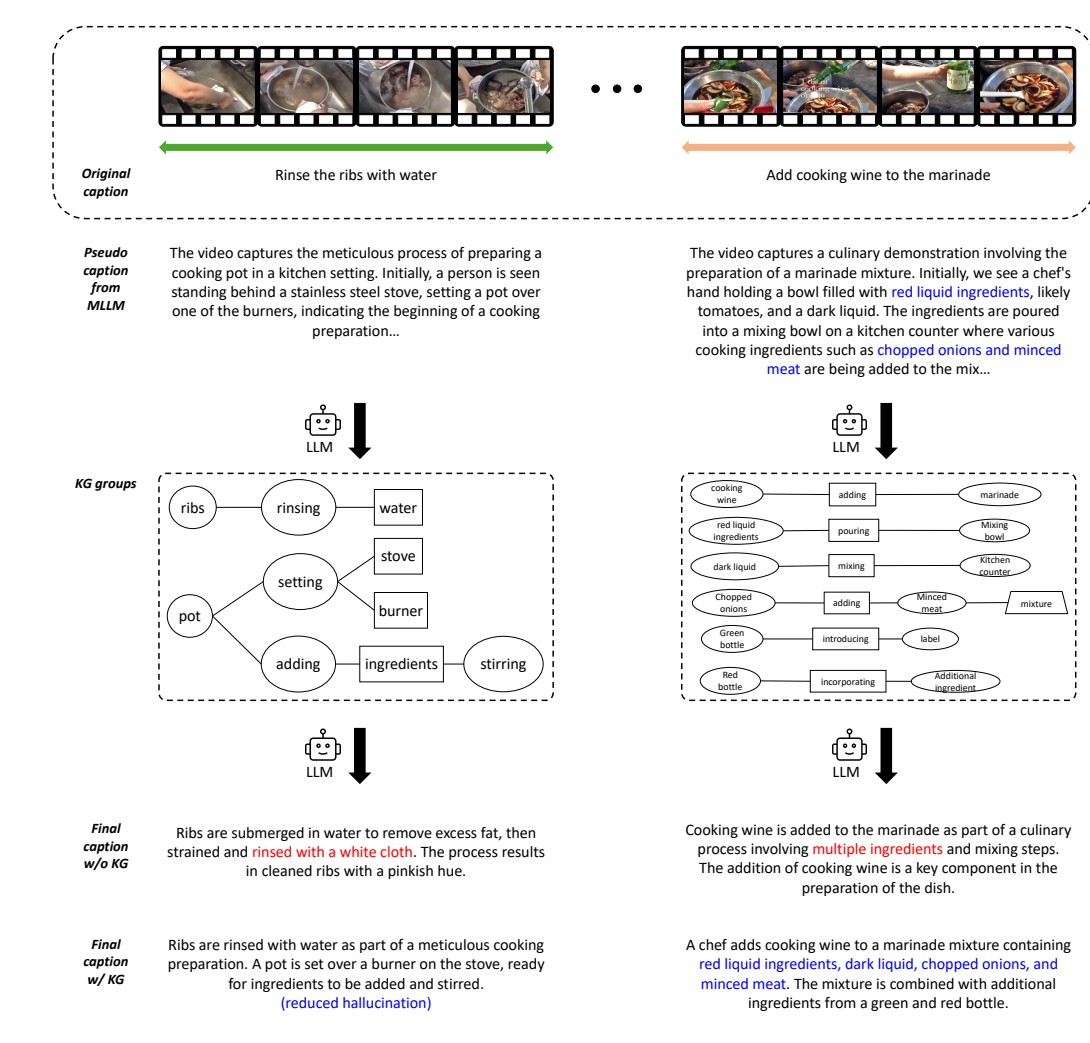

Figure 11: Visualization Results of Knowledge Graph.

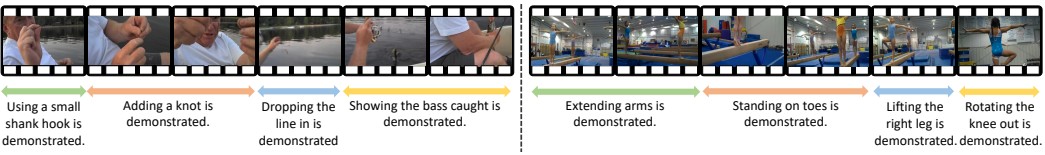

Figure 12: Visualization Results of General Domain Videos.

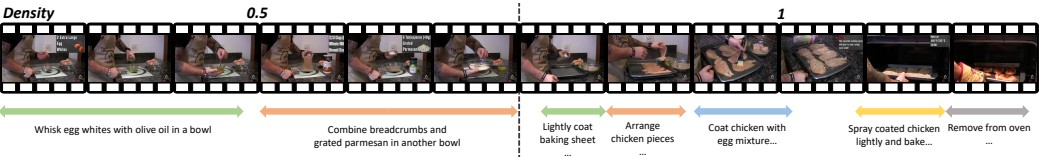

Figure 13: Visualization Results for streaming scenario on the YouCook2 *val* set.

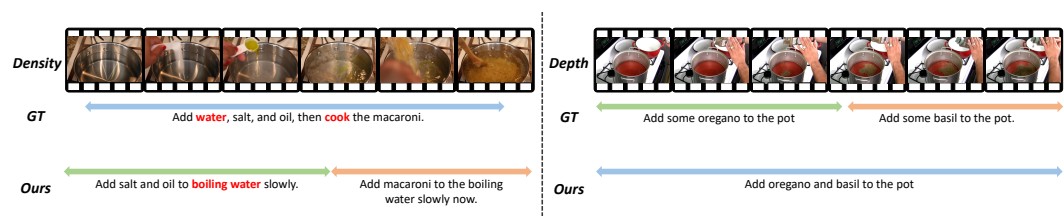

Figure 14: Visualization Results of failure cases on YouCook2 *val* set.

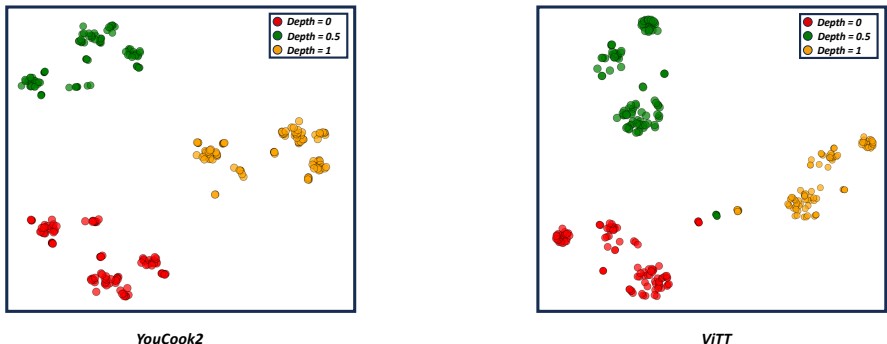

Figure 15: t-SNE visualization of depth encoder features for depth values 0.0, 0.5, and 1.0.

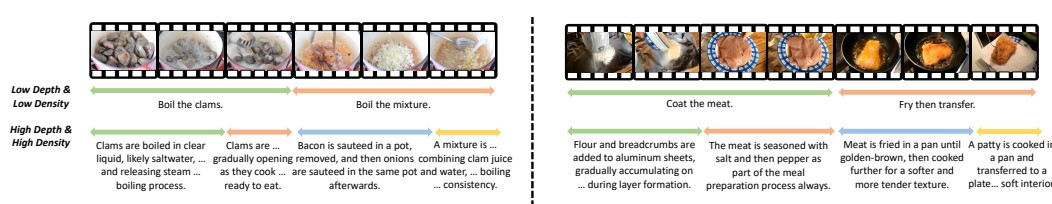

Figure 16: Visualization of difficult scenarios in UC-Captions.

