# OpenReview forum: "User-Controllable Dense Video Captioning: A Large-Scale Benchmark and Framework"
_ICLR.cc/2026/Conference — Submitted to ICLR 2026_

### Official Review · Reviewer_e2hm · 2025-10-30

**Soundness:** 3
**Presentation:** 4
**Contribution:** 2
**Rating:** 4
**Confidence:** 3

**Summary:**

This paper introduces UC Captions, a benchmark for user-controllable dense video captioning (DVC), and proposes the UC-DVC framework, which allows users to specify both event density and caption depth. The dataset extends existing DVC benchmarks with multi-level, continuous annotations, and the model incorporates user control via dedicated modules. Experiments show strong results on instructional video datasets.

**Strengths:**

1. System integration: The benchmark and framework are well-integrated and address a practical need for user-controllable video captioning.
2. Experimental thoroughness (within DVC): The experiments are comprehensive for the DVC task, including ablations and human evaluations.
3. Clarity: The paper is well-written and the methodology is transparent.

**Weaknesses:**

1. Motivation unclear: The practical necessity and real-world demand for user-controllable dense captioning are not convincingly established. The paper lacks user studies, deployment evidence, or concrete scenarios demonstrating that end-users benefit from or desire explicit control over event density and caption depth. The usability of such controls for non-expert users is not discussed, and it is unclear how these parameters would be presented or utilized in real applications.
2. Limited generalization: All experiments are restricted to instructional video datasets and the DVC task. There is no empirical evidence that the benchmark or model benefits other video understanding tasks (e.g., event detection, VQA, retrieval, temporal segmentation).
3. Algorithmic novelty: The modeling and data generation techniques are incremental adaptations of existing ideas, not fundamentally new algorithms.
4. Synthetic data risks: Heavy reliance on synthetic captions and knowledge graphs may introduce distributional artifacts or biases, which are not analyzed or addressed.
5. Lack of robustness Analysis: The model’s robustness to noisy, ambiguous, or adversarial user controls is not evaluated.
6. Unsubstantiated claims: The paper claims flexibility and broad applicability, but provides no experiments or case studies beyond DVC to support these assertions.

**Questions:**

1. Motivation and usability: Can the authors provide evidence (e.g., user studies, deployment feedback, or real-world use cases) that explicit user control over event density and caption depth is needed or beneficial in practice? How would non-expert users interact with these controls in real applications?
2. Broader applicability: Can the authors provide empirical evidence or at least a concrete case study showing how UC Captions or UC-DVC can benefit tasks beyond video captioning? For example, can the benchmark be used for event detection, VQA, or video retrieval? If not, the claims of generality should be toned down.
3. Synthetic data bias: What steps have been taken to identify and mitigate biases or artifacts introduced by synthetic data and knowledge graphs? Are there measurable differences between synthetic and human-annotated captions?
4. Robustness to user input: How does the model handle ambiguous, conflicting, or adversarial user controls? Are there failure cases?
5. Comparison to other paradigms: How does UC-DVC compare to prompt-based or reinforcement learning-based controllable generation frameworks, both in flexibility and performance?
6. Generalization to other domains: Have the authors attempted to apply the framework to non-instructional videos or more diverse datasets?

---

> ### Author Response · Authors · 2025-11-21
> **Response to Reviewer e2hm [1/4]**
>
> >W1: **Motivation unclear:** The practical necessity and real-world demand for user-controllable dense captioning are not convincingly established. The paper lacks user studies, deployment evidence, or concrete scenarios demonstrating that end-users benefit from or desire explicit control over event density and caption depth. The usability of such controls for non-expert users is not discussed, and it is unclear how these parameters would be presented or utilized in real applications.
>
> >Q1: **Motivation and usability:** Can the authors provide evidence (e.g., user studies, deployment feedback, or real-world use cases) that explicit user control over event density and caption depth is needed or beneficial in practice? How would non-expert users interact with these controls in real applications?
>
> Answer: We thank the Reviewer for raising these fundamental questions regarding practical motivation. We address the necessity of our framework from two perspectives: specific end-user demands and domain expert consensus.
>
> First, **regarding non-expert users**, HCI research confirms that user needs are context-dependent. Jiang et al. (CHI 2024) [b] report that Blind and Low Vision users prefer **detailed descriptions for education but concise ones for entertainment.** Our $(d, z)$ control provides the technical mechanism to satisfy these conflicting demands within a single model. Regarding usability, these parameters can be mapped to intuitive presets (e.g., "Summary Mode" vs. "Study Mode") in real-world applications to ensure seamless interaction.
>
> Second, **regarding domain experts**, we respectfully highlight that **other reviewers have strongly affirmed the practical necessity of our approach.** Reviewer 1 (fkPC) agreed that "different video applications focus on varying levels of semantic granularity" and deemed our motivation "natural and reasonable." Similarly, Reviewer 2 (QKKP) stated that our control mechanism "addresses practical needs, ranging from fast summaries to detailed descriptions," and Reviewer 3 (4Kqq) recognized that "the motivation is clearly articulated" and "could be useful for future research." This alignment between empirical user needs and expert assessment confirms that User-Controllable DVC is a well-motivated and necessary direction.
>
> [b] Jiang, Lucy, et al. "It's Kind of Context Dependent": Understanding Blind and Low Vision People's Video Accessibility Preferences Across Viewing Scenarios, in CHI 2024.
>
> >W2: **Limited generalization:** All experiments are restricted to instructional video datasets and the DVC task. There is no empirical evidence that the benchmark or model benefits other video understanding tasks (e.g., event detection, VQA, retrieval, temporal segmentation).
>
> >Q2: **Broader applicability:** Can the authors provide empirical evidence or at least a concrete case study showing how UC Captions or UC-DVC can benefit tasks beyond video captioning? For example, can the benchmark be used for event detection, VQA, or video retrieval? If not, the claims of generality should be toned down.
>
> >W6: **Unsubstantiated claims:** The paper claims flexibility and broad applicability, but provides no experiments or case studies beyond DVC to support these assertions.
>
> Answer: We thank the Reviewer for this fair assessment. We agree that our claims regarding broad applicability across general video understanding tasks (e.g., VQA, retrieval, event detection) were overstated given that our empirical evidence is centered on Dense Video Captioning (DVC). We have **revised the manuscript to tone down these assertions** and explicitly frame broader applications as future directions rather than current contributions.
>
> Specifically, in the Conclusion, we **modified the statement** "hold practical applications across diverse video understanding tasks" to "**potentially serve** as a useful resource for future work exploring other video understanding tasks," thereby aligning our claims strictly with our experimental results.
>
> Furthermore, regarding the Reviewer's concern about **"unsubstantiated claims of flexibility"** we clarify that our intended scope of "flexibility" and "applicability" refers **specifically to the granular control within the DVC task**—namely, the novel ability to dynamically adjust event density and caption depth to meet diverse user needs (e.g., from concise summaries to detailed narratives). We believe this revision accurately reflects our core contribution while maintaining scientific rigor.

---

> ### Author Response · Authors · 2025-11-21
> **Response to Reviewer e2hm [2/4]**
>
> >W3: **Algorithmic novelty:** The modeling and data generation techniques are incremental adaptations of existing ideas, not fundamentally new algorithms.
>
> Answer: We thank the Reviewer for the feedback regarding algorithmic novelty, which allows us to clarify the specific methodological contributions of our work. We emphasize that our work contains algorithmic novelty **specifically tailored to solve the unexplored problem of user-controllable dense video captioning.** This novelty arises in three key components designed to address the unique challenges of multi-granularity generation.
>
> First, we introduce a **GT-anchored synthetic data algorithm for density control**, which fundamentally differs from standard sliding-window augmentation [c] by preserving the semantic structure of the original Ground Truth while expanding event density.
>
> Second, we propose a **TKG-based caption generation pipeline for depth control**; unlike generic LLM post-editing, our method utilizes graph-grounding to structurally constrain the model, effectively preventing hallucination during depth expansion.
>
> Third, the **UC-DVC framework** itself represents a novel architectural integration, combining continuous control embeddings with specialized density- and depth-aware losses to successfully manage multi-granularity supervision signals within a single unified model. We believe these carefully designed components collectively constitute a significant methodological advancement that enables precise user control in DVC for the first time.
>
> [c] Chen, Lin, et al. "Sharegpt4video: Improving video understanding and generation with better captions.", in NeurIPS 2024.
>
> >W4: **Synthetic data risks:** Heavy reliance on synthetic captions and knowledge graphs may introduce distributional artifacts or biases, which are not analyzed or addressed.
>
> >Q3: **Synthetic data bias:** What steps have been taken to identify and mitigate biases or artifacts introduced by synthetic data and knowledge graphs? Are there measurable differences between synthetic and human-annotated captions?
>
> Answer: We thank the Reviewer for raising this critical point regarding the potential risks of synthetic data generation and the need for rigorous quality control.
>
> First, regarding the steps taken to identify and mitigate biases or artifacts, we implemented a GT-anchored **Textual Knowledge Graph (TKG)** strategy. This approach **explicitly anchors the synthetic captions to the original human Ground Truth**, preventing the model from drifting into unnatural distributions or generating artifacts. As illustrated in the qualitative comparison in Figure 11, generating captions **without the TKG can lead to hallucinations**; for instance, the model incorrectly invents that ribs were "rinsed with a white cloth". In contrast, by instructing the model to elaborate only on entities and actions explicitly present in the KG (e.g., "pot," "rinsing," "water"), the TKG-guided output accurately generates "Ribs are rinsed with water as part of a meticulous cooking…" effectively reducing hallucination while maintaining descriptive richness. The effectiveness of this design is further supported by our ablation study (Table 4), which shows a clear **performance drop when the TKG is removed.**
>
> Second, regarding whether there are measurable differences between synthetic and human-annotated captions, we empirically investigated this through the extensive **Human Evaluation** reported in Appendix D. We recruited 14 experts to assess 504 randomly sampled video-caption pairs. The results in Table 12 confirmed that the quality of **our synthetic captions is comparable to human-annotated ground truth** (e.g., caption quality scores of 2.52--2.93 for synthetic vs. 2.93 for human anchors on YouCook2). These findings serve as empirical evidence that our synthetic data does not exhibit significant negative biases or distributional artifacts compared to human annotations.

---

> ### Author Response · Authors · 2025-11-21
> **Response to Reviewer e2hm [3/4]**
>
> >W5: **Lack of robustness analysis:** The model's robustness to noisy, ambiguous, or adversarial user controls is not evaluated.
>
> >Q4: **Robustness to user input:** How does the model handle ambiguous, conflicting, or adversarial user controls? Are there failure cases?
>
> | Input Value | Depth Effect (Avg. Words) | Density Effect (Total Events) |
> | :---: | :---: | :---: |
> | 0.40 | 7.71 | 3,251 |
> | **0.50 (Base)** | **8.33** | **3,383** |
> | 0.51 | 8.73 | 3,422 |
> | 0.60 | 10.44 | 3,555 |
> | 0.75 | 13.79 | 3,793 |
>
> Answer: We thank the Reviewer for this important question regarding the robustness of the model. Among the scenarios suggested by the Reviewer, we specifically address **noisy** and **conflicting input** cases as representative examples below. Additionally, we have provided an analysis of **failure cases** in Appendix G (Figure 14).
>
> **1. Robustness to Noisy Inputs:** To evaluate robustness against input noise, we conducted a sensitivity analysis by introducing varying degrees of noise to the control inputs around a base value of 0.5. We measured the change in Caption Length (Avg. Words) when varying Depth (fixing Density at 0.5) and Event Count (Total Timestamps) when varying Density (fixing Depth at 0.5). As observed, **our model exhibits locally smooth and stable behavior.** For instance, a minor noise of 0.01 (from 0.50 to 0.51) results in a marginal and proportional increase in both caption length (8.33 to 8.73) and event counts (3,383 to 3,422). Even with larger deviations (0.40 to 0.75), the output metrics change in a nearly linear and gradual manner.
>
> **2. Handling Conflicting Inputs:** Regarding "conflicting" inputs (e.g., specifying Low-Density with Low-Depth or vice versa), our framework is explicitly designed to handle such combinations. As shown in Table 3 of the revised manuscript, we evaluated the model on all 9 combinations ($3 \times 3$ spectrum) of density and depth. The results confirm that **UC-DVC consistently outperforms baselines** even in these **corner cases**, effectively disentangling density and depth requirements to generate appropriate captions.
>
> **3. Analysis of Failure Cases:** As illustrated in Appendix G, Figure 14, we analyzed representative error modes encountered by our framework. Specifically, we observe instances of over-segmentation with hallucination (incorrectly fragmenting continuous events) and under-segmentation (merging distinct atomic actions due to visual similarity). These examples highlight current limitations in maintaining physical state consistency and distinguishing fine-grained actions in rapid succession.
>
> >Q5: **Comparison to other paradigms:** How does UC-DVC compare to prompt-based or reinforcement learning-based controllable generation frameworks, both in flexibility and performance?
>
> | Method | METEOR | CIDEr | SODA_c | F1 |
> | :--- | :---: | :---: | :---: | :---: |
> | VideoExpert [c] | 4.80 | 18.70 | 4.20 | - |
> | Ours (Discrete Prompt) | 12.41 | 69.01 | 10.55 | 32.46 |
> | **Ours (Continuous)** | **12.92** | **75.97** | **11.91** | **37.90** |
>
> Answer: We thank the Reviewer for this insightful question. As per the Reviewer's request to compare with other paradigms, we conducted a comprehensive analysis against the prompt-based generation framework.
>
> First, to evaluate a representative prompt-based framework within our architecture, we conducted a new experiment where we **replaced our continuous scalar inputs ($d, z \in [0,1]$) with prompt-based inputs (e.g., "Low", "Mid", "High").** This ensures a direct comparison of the control mechanism itself while keeping the rest of the model identical. Second, we additionally compared our method with VideoExpert [c], a recent general-purpose prompt-based MLLM, to assess performance against existing baselines. We report the comparative results in the table above.
>
> As shown in the table, while our prompt-based variant yields reasonable results (69.01 CIDEr), it consistently underperforms our proposed continuous control method (75.97 CIDEr). This confirms that **our continuous embedding mechanism offers superior optimization dynamics and more precise control** than prompt-based conditioning. Furthermore, the performance gap is even more pronounced when compared to the general-purpose MLLM, VideoExpert, which achieves a CIDEr score of only 18.7. These comparisons collectively highlight that our continuous control mechanism provides a fundamentally richer and more effective representation for dense video captioning than prompt-based alternatives.
>
> [c] Zhao, Henghao, et al. "VideoExpert: Augmented LLM for Temporal-Sensitive Video Understanding.", in arXiv 2025.

---

> ### Author Response · Authors · 2025-11-21
> **Response to Reviewer e2hm [4/4]**
>
> >Q6: **Generalization to other domains:** Have the authors attempted to apply the framework to non-instructional videos or more diverse datasets?
>
> Answer: We thank the Reviewer for this important question regarding the generalization of our framework to non-instructional domains. To address this, we have conducted a qualitative analysis on **general domain videos** retrieved from YouTube and included the results in Appendix G (Figure 12) of the revised manuscript.
>
> As illustrated in the figure, our framework demonstrates robust generalization capabilities beyond the culinary focus of the training data. For instance, in **outdoor activity scenarios** like fishing, the model **accurately describes procedural steps** such as "Adding a knot" and "Dropping the line in". Similarly, in **physical exercise videos**, it **precisely captures fine-grained body movements** like "Standing on toes" and "Rotating the knee out". These results confirm that our user-controllable dense captioning capability behaves intuitively and effectively transfers to diverse, non-instructional settings involving complex temporal events.

---

### Official Review · Reviewer_4Kqq · 2025-10-30

**Soundness:** 3
**Presentation:** 3
**Contribution:** 3
**Rating:** 4
**Confidence:** 3

**Summary:**

The paper identifies a key limitation of prior dense video captioning (DVC) systems: existing benchmarks provide only a single annotation style, so models can produce only fixed captions without user control over the level of detail.
To address this gap, the authors introduce UC Captions, a new dataset that provides captions with controllable variation along two dimensions: event density (how many events are detected) and caption depth (how detailed the descriptions are). By adjusting these dimensions, users can obtain captions of different granularity.
The authors also propose UC-DVC, a framework that conditions caption generation on user-specified density and depth parameters.

**Strengths:**

- The motivation is clearly articulated: enabling controllable video captioning along two key dimensions—event density and caption depth.
- The introduction of the UC Captions dataset, which provides multi-level annotations for both density and depth, could be useful for future research in this area.
- Experiments show that UC-DVC performs strongly on standard DVC benchmarks and adapts well across multiple granularity configurations.
- The paper includes thorough analyses of how varying density and depth affect model outputs.

**Weaknesses:**

- It is unclear whether caption depth is truly reflected by surface measures such as word count. The paper does not provide an analysis on whether depth labels correlate with quantitative metrics (e.g., length, vocabulary richness), nor whether “deeper” captions provide meaningfully more information rather than just being longer.
- The dataset construction involves automated filtering and refinement, but the paper provides limited discussion on quality control. It is not clear how the authors verify that the revised density and depth annotations are correct. In particular, for the evaluation in Table 3, it appears that the models are evaluated on the newly constructed dataset, where it is also unclear how the filtering process and quality are controlled.

**Questions:**

- For density modeling, the MLP layer appears to take only d_in as input. If it is not conditioned on video features, wouldn’t this produce the same transformation for all videos?
- In the training data, are the ground-truth control values limited to only three discrete levels (0, 0.5, 1)?
- In Table 2, what does “our environment” refer to? What changes have you made from the original models?
- The paper states that prior work lacks mechanisms to model variation in density and depth; however, it is still unclear why performance drops when training with UC Captions. Do you have intuition or analysis explaining this decline?
- Event density and caption depth seem inherently related. Did you observe interactions between the two? For example, when density is high, it seems to be more difficult to maintain high caption depth, which could potentially increase hallucination.

---

> ### Author Response · Authors · 2025-11-21
> **Response to Reviewer 4Kqq [1/3]**
>
> >W1: It is unclear whether caption depth is truly reflected by surface measures such as word count. The paper does not provide an analysis on whether depth labels correlate with quantitative metrics (e.g., length, vocabulary richness), nor whether "deeper" captions provide meaningfully more information rather than just being longer.
>
> Answer: We thank the Reviewer for this critical question. We agree that establishing 'depth' as a meaningful concept beyond simple 'word count' is essential. To address this, we have conducted a two-fold analysis focusing on lexical and semantic richness, and we have included these results in the "More Analysis in UC-Captions" paragraph within the Discussion subsection of Section 4 (Experiments) in the revised manuscript.
>
> First, we demonstrate that our depth control induces a fundamental expansion in vocabulary usage. We analyzed the number of **Unique Tokens**—defined as the count of distinct word types used within the caption set—as a metric for lexical diversity. Our results confirm that **deeper captions are built from a systematically larger vocabulary pool**; for instance, in the YouCook2 dataset, the High depth level utilizes 4,730 unique tokens, representing a 4.5 times increase over the 1,058 tokens used at the Low depth level. This proves that the model accesses a significantly richer lexicon at higher depth levels rather than merely generating longer sentences.
>
> Second, to determine if this richness conveys meaningfully more information (i.e., semantic information), we employed a frontier-level LLM (Gemini-2.5-Flash) as an evaluator [a]. We tasked the model with assessing **"Semantic Richness"** (0-1 scale) and ranking the captions by informativeness. The findings reveal a strong **monotonic alignment between our depth levels and semantic value.** For example, on YouCook2, the average richness score rises significantly from 0.37 (Low) to 0.95 (High), and the High-depth captions were consistently ranked as the best (Average Rank 1.10). These combined analyses validate that our depth control serves as a **robust mechanism for modulating both lexical diversity and semantic information content.**
>
> [a] Zheng et al., "Judging LLM-as-a-Judge with MT-Bench and Chatbot Arena," NeurIPS 2023.
>
> >W2: The dataset construction involves automated filtering and refinement, but the paper provides limited discussion on quality control. It is not clear how the authors verify that the revised density and depth annotations are correct. In particular, for the evaluation in Table 3, it appears that the models are evaluated on the newly constructed dataset, where it is also unclear how the filtering process and quality are controlled.
>
> Answer: We thank the Reviewer for this critical question regarding the verification of our automated refinement process. We fully agree that ensuring the correctness of the revised annotations is essential for the reliability of our evaluation. We address this through a two-step validation process: empirical verification via **human evaluation and methodological quality control via our TKG approach**.
>
> First, regarding the verification that the revised annotations are correct, we would like to highlight that we empirically validated this through the extensive Human Evaluation reported in Appendix D. To verify the correctness of the constructed dataset, we recruited 14 experts to assess 504 randomly sampled video-caption pairs. The results confirmed that **our automated density filtering is highly accurate** (e.g., 97.6% accuracy for ViTT High-to-Low control) and that the **caption quality is comparable to human-annotated ground truth.** These results serve as empirical evidence that the revised density and depth annotations are correct and align with human perception.
>
> Second, regarding the control of the filtering process and generation quality, we implemented a GT-anchored **Textual Knowledge Graph (TKG)** strategy to structurally guarantee accuracy and mitigate hallucinations. As demonstrated in the qualitative analysis in Figure 11, the **absence of the TKG constraint can result in factual errors**; for example, the model erroneously described ribs as being "rinsed with a white cloth". Conversely, by constraining the model to expand solely upon entities and actions verified within the KG (e.g., "pot," "rinsing," "water"), the TKG-driven generation yields precise outputs such as "Ribs are rinsed with water as part of a meticulous cooking…", thereby effectively curtailing hallucination while preserving descriptive detail. The validity of this approach is further corroborated by our ablation study (Table 4), where **removing the TKG component resulted in a distinct decline in performance.**

---

> ### Author Response · Authors · 2025-11-21
> **Response to Reviewer 4Kqq [2/3]**
>
> >Q1: For density modeling, the MLP layer appears to take only d_in as input. If it is not conditioned on video features, wouldn’t this produce the same transformation for all videos?
>
> Answer: We thank the Reviewer for this insightful question regarding the conditioning mechanism of our density modeling layer. We fully agree with the Reviewer's core intuition; if the **density input were not conditioned** on the video features, it would indeed result in the same problematic transformation for all videos. However, **our framework is explicitly designed to be content-aware.** As shown in Eq. (2) and Eq. (3), the scalar density input $d_{in}$ is first projected to an embedding $e_d$. This **embedding is then combined with the visual features $V$** by concatenation before being fed into the density encoder $E_d$. Because $E_d$ operates on $[V || e_d]$, the **resulting transformation is jointly conditioned on both the density input and the video features**, not a single global mapping.
>
> >Q2: In the training data, are the ground-truth control values limited to only three discrete levels (0, 0.5, 1)?
>
> Answer: Yes, during training, the ground-truth supervision is indeed provided at **three discrete anchor levels: $\{0.0, 0.5, 1.0\}$.** We employ these discrete anchors because they provide clear, well-defined supervision signals that establish a structured control space for the model to learn from. However, it is important to note that our control mechanism, which **utilizes continuous MLPs, is explicitly designed to model a continuous control space** rather than a categorical one. Consequently, although the model is trained on these discrete anchor points, it naturally generalizes to intermediate values (e.g., 0.25, 0.75) at inference time. We empirically validate this in Figure 9 and Figure 10 (Appendix G), which demonstrates that sweeping the control values produces smooth and gradual changes in event counts and caption lengths, confirming the ability of the model to handle continuous interpolation.
>
> >Q3: In Table 2, what does “our environment” refer to? What changes have you made from the original models?
>
> Answer: We thank the Reviewer for requesting clarification on the term "our environment" used in Table 2. "Our environment" means that codes were **re-implemented with official code and trained under our experimental setting.** Concretely, this includes using the same video encoder backbone, input resolution, optimizer, batch size, and number of training epochs to ensure a fair comparison.
>
> >Q4: The paper states that prior work lacks mechanisms to model variation in density and depth; however, it is still unclear why performance drops when training with UC Captions. Do you have intuition or analysis explaining this decline?
>
> Answer: We thank the Reviewer for this insightful question, requesting the intuition behind the baseline performance drop when using UC Captions. The fundamental reason lies in the supervision signals. **UC Captions**, by design, contains captions with **different event densities and depths for the same video input**. Therefore, during training, we randomly select one of those different captions with densities and depths. However, **prior models do not explicitly distinguish these levels**—indeed, existing methods have never incorporated any control mechanism for density or depth—and therefore treat all these diverse variants as a single, homogeneous supervision signal. This creates conflicting optimization where the model is forced to simultaneously minimize loss for contradictory targets (e.g., short vs. long captions) without any distinguishing input. This ambiguity confuses the model, leading to the observed performance drop. In contrast, **UC-DVC** is explicitly conditioned on the control parameters $(d, z)$, allowing it to **learn exactly "which kind of caption should be produced at which level"**, thus systematically exploiting the mixed granularities rather than being confused by them.

---

> ### Author Response · Authors · 2025-11-21
> **Response to Reviewer 4Kqq [3/3]**
>
> >Q5: Event density and caption depth seem inherently related. Did you observe interactions between the two? For example, when density is high, it seems to be more difficult to maintain high caption depth, which could potentially increase hallucination.
>
> Answer: We thank the Reviewer for this insightful question regarding the inherent interaction between density and depth, and the potential risk of hallucination. We address this via both our method and our data generation.
>
> (1) **Method Perspective**: We use specialized losses. The **density-aware loss encourages clear boundary separation**, even when density is high. The **depth-aware loss uses saliency-token supervision to ensure "depth" means adding salient**, GT-grounded tokens, not just arbitrary words. Empirically, as shown in Table 3, our model maintains robust performance across all 9 density-depth combinations. Notably, even in the most challenging setting of (High Density, High Depth), our model achieves high METEOR and F1 scores (e.g., 9.1 and 29.9 on YouCook2), outperforming the baseline and demonstrating that it can successfully handle high-density events without compromising caption quality.
>
> (2) **Data-Generation Perspective**: Our pipeline is **TKG-based.** We instruct the LLM to expand depth only using entities and relations present in the Knowledge Graph. This grounds the high-depth captions in the GT/TKG, **structurally preventing the model from inventing new objects or actions** (hallucination). Furthermore, we do not enforce a fixed word count for high-depth captions; instead, we provide a **flexible target range (e.g., 18-20 words).** This design choice prevents the LLM from being forced to generate superfluous or ungrounded content simply to meet a rigid length requirement, which provides an additional safeguard against hallucination. The success of these measures is further **validated by our Human Evaluation** (Appendix D, Table 12). The results confirm that our generated captions achieve high appropriateness scores comparable to human ground-truth across all levels (e.g., 2.52--2.93 vs. Human 2.93 on YouCook2), indicating that increasing depth maintains factual accuracy rather than introducing hallucinations.

---

> > ### Comment · Reviewer_4Kqq · 2025-11-27
> >
> > Thank you for your detailed response. I have a few remaining questions:
> > W1. Could you also share the prompt related to semantic richness? I assume that a longer context contains more information, but I’m not sure whether that additional information is actually meaningful.
> > Q1. If that is the case, what is the purpose of the projection? Wouldn’t it always produce the same result for the same input?
> > Q2. I’m still not fully convinced how a discrete training dataset can generalize to continuous inputs at inference time. Could you provide a more detailed example, for instance, how the model interprets inputs like 0.5 vs. 0.6?
> > Q4. In the current experiment, are you training with the same input but different outputs? What would happen if we included event density and depth information directly in the input so that different outputs correspond to different inputs?
> > Q5. Could you share some examples that illustrate high density with high caption depth, and vice versa?

---

> ### Author Response · Authors · 2025-12-01
> **Response to Remaining Questions for Reviewer 4Kqq [1/2]**
>
> > W1: Could you also share the prompt related to semantic richness? I assume that a longer context contains more information, but I’m not sure whether that additional information is actually meaningful.
>
> Answer: We thank the Reviewer for the question regarding the evaluation of semantic richness. To address this concern, **we share the exact prompts used for our LLM-based evaluation** below. As you rightly pointed out, longer context does not inherently guarantee meaningful information. To address this, our evaluation protocol explicitly defines 'Semantic Depth' not by **simple word count**, but by **the density of specific details** such as the agent, action, and tool. The prompt instructs the evaluator to rate and rank captions based on this definition of richness. The prompts used for the YouCook2 dataset are as follows:
>
> **Scoring Prompt (YouCook2)**
>
> ```
> def get_scoring_yc2_prompt(captions):
>     return f"""You are an expert evaluator for COOKING video captions. Rate 'Semantic Depth' (0.0-1.0).
> ### Definition
> Depth = Density of specific cooking details (Agent, Action, Ingredient, Tool, Context).
> ### Definition of 'Semantic Depth'
> 'Semantic Depth' is NOT about simple word count. It is defined by the density of specific cooking details. Evaluate based on the richness of details regarding:
> 1. [Agent]: Who (e.g., "a chef" > "a person")
> 2. [Action]: How (e.g., "finely chops" > "cuts")
> 3. [Ingredient/Object]: What (e.g., "fresh basil" > "herbs")
> 4. [Tool/Context]: Where/With what (e.g., "in a non-stick pan" > "in a pan")
> ### Task
> Score these three captions based on the definition above.
> ### Rubric
> - 0.0: "A person is shown." (Vague)
> - 0.5: "A woman cuts vegetables." (Basic action)
> - 1.0: "A woman in a red shirt slices carrots on a wooden board." (Detailed)
> Input Captions:
> A: "{captions[0][1]}"
> B: "{captions[1][1]}"
> C: "{captions[2][1]}"
> Respond JSON: {{ "A": 0.x, "B": 0.x, "C": 0.x }}"""
> ```
>
> **Ranking Prompt (YouCook2)**
>
> ```
> def get_yc2_ranking_prompt(captions):
>     return f"""You are an expert evaluator for COOKING video captions. Your task is to rank the following three captions describing the same event from LEAST Semantic Depth to MOST Semantic Depth.
> ### Definition of 'Semantic Depth'
> 'Semantic Depth' is NOT about simple word count. It is defined by the density of specific cooking details. Evaluate based on the richness of details regarding:
> 1. [Agent]: Who (e.g., "a chef" > "a person")
> 2. [Action]: How (e.g., "finely chops" > "cuts")
> 3. [Ingredient/Object]: What (e.g., "fresh basil" > "herbs")
> 4. [Tool/Context]: Where/With what (e.g., "in a non-stick pan" > "in a pan")
> ---
> ### Task
> Rank these three captions based on the definition above.
> Captions:
> A: "{captions[0][1]}"
> B: "{captions[1][1]}"
> C: "{captions[2][1]}"
> Respond with a JSON object containing the list of letters in order (Least -> Middle -> Most):
> {{ "ranking_order": ["Letter_Least", "Letter_Middle", "Letter_Most"] }}"""
> ```
>
> These prompts ensure that the **'Semantic Richness' score** reflects the qualitative increase in information content, which aligns with our findings in Table 6 where high-depth captions achieved superior semantic scores (0.37 -> 0.95)  and rankings (Avg. 1.10).
>
> > Q1: If that is the case, what is the purpose of the projection? Wouldn’t it always produce the same result for the same input?
>
> Answer: We thank the Reviewer for the question. We intentionally design the MLP layer to produce the same transformation for the same scalar input $d_{in}$, regardless of video content. Its purpose is to project the low-dimensional scalar input (e.g., $d_{in} \in [0, 1]$) into a high-dimensional embedding space that acts as a **global conditioning signal** compatible with the video feature dimension.
> This is similar to \<$cls$\> token in ViT that are projected and injected to provide global signals. Although the projection itself is content-independent, the subsequent Density Encoder processes this embedding jointly with the video features to modulate event localization based on the projected density level.

---

> ### Author Response · Authors · 2025-12-01
> **Response to Remaining Questions for Reviewer 4Kqq [2/2]**
>
> > Q2: I’m still not fully convinced how a discrete training dataset can generalize to continuous inputs at inference time. Could you provide a more detailed example, for instance, how the model interprets inputs like 0.5 vs. 0.6?
>
> Answer: We thank the reviewer for the question. The key reason is the continuity of the MLP, which provides a smooth mapping from the scalar control value to the density embedding. Because an MLP is composed of affine transformations and continuous activation functions, it implements a continuous function over the input domain [d]. Thus, even though the model is trained only on discrete anchor points $\{0.0, 0.5, 1.0\}$, it learns a single smooth function that **naturally interpolates intermediate values** such as 0.6.
>
> In practice, the model interprets 0.6 as a slight increase from 0.5. Locally, this can be approximated by $e_d(0.6) \approx e_d(0.5) + J\big|_{x=0.5} \cdot \Delta x, \quad \Delta x = 0.1,$ ($J\big|\_{x=0.5}$ is the Jacobian of the MLP at $ x = 0.5 $). This indicates that the embedding for 0.6 lies on a linear trajectory in the latent space between the embeddings of 0.5 and 1.0, rather than forming an isolated category.
>
> Empirically, as shown in the below table, sweeping the input control value around (e.g., from $0.5$ to $0.6$) produces **smooth and nearly linear** changes in output. This confirms that the model **generalizes to intermediate control values** during training.
>  | Input Value | Depth Effect (Avg. Words) | Density Effect (Total Events) |
> | :---: | :---: | :---: |
> | 0.40 | 7.71 | 3,251 |
> | **0.50 (Base)** | **8.33** | **3,383** |
> | 0.51  | 8.73 | 3,422 |
> | 0.60 | 10.44 | 3,555 |
> | 0.75 | 13.79 | 3,793 |
>
> [d] K. Hornik, "Approximation Capabilities of Multilayer Feedforward Networks," Neural Networks 1991.
>
> > Q4: In the current experiment, are you training with the same input but different outputs? What would happen if we included event density and depth information directly in the input so that different outputs correspond to different inputs?
>
> Answer: We thank the Reviewer for this insightful question. For the first question, the answer is “yes”: in the baseline experiments, the models are trained with the same video input but different target outputs. Please note that, as standard DVC models take only **Video and Speech Transcripts as input**, they fundamentally lack a mechanism to distinguish varying targets. This leads to conflicting supervision and the performance decline (Table 2).
>
> For the second question, we fully agree with your intuition that including density and depth information directly in the input. Technically, while it is possible to merge signals into features, our goal of this manuscript was to implement this with **minimal architectural modifications** while **maintaining the standard DVC form** for the flexible output modulation. To this end, we employed an MLP projection to seamlessly integrate **user-friendly scalar controls ($d, z$)** into the video features. This design effectively transforms the problem into a well-defined mapping. Table 5 empirically demonstrates that providing these user-friendly inputs enables the model to effectively distinguish between the 9 granularity levels, and Table 14 confirms that our specialized losses are necessary to guide the model in utilizing these signals.
>
> > Q5: Could you share some examples that illustrate high density with high caption depth, and vice versa?
>
> Answer: We thank the Reviewer for the question. We have provided the mentioned examples in **Figure 16 (Appendix G)** of the revised manuscript. These examples confirm that UC Captions properly reflects the intended density and caption depth, providing high-quality annotations without hallucination.

---

### Official Review · Reviewer_QKKP · 2025-11-01

**Soundness:** 3
**Presentation:** 3
**Contribution:** 3
**Rating:** 6
**Confidence:** 4

**Summary:**

The paper introduces User-Controllable Dense Video Captioning (UC-DVC) and a companion dataset, UC Captions, to let users control (i) event density (how many events are detected) and (ii) caption depth (how detailed each description is). The dataset expands YouCook2 and ViTT with a 3×3 grid of density×depth annotations via an automated pipeline (visual-similarity–based splitting/merging and a textual knowledge graph for multi-depth captions). The UC-DVC model adds explicit density and depth encoders plus two new losses (density-aware, depth-aware) to modulate event localization and caption detail. On YouCook2/ViTT, the approach improves METEOR/CIDEr/SODA c/F1 over strong baselines and, unlike baselines, covers all nine granularity settings with a single model.

**Strengths:**

1. User control over density and depth addresses practical needs, ranging from fast summaries to detailed descriptions. The paper provides a well-motivated analysis of this gap.

2. A principled pipeline is proposed that defines levels and refines density through Clip Inter-/Intra-Median thresholds and MaxDiv (i.e., action count derived from captions), while depth is refined using a Textual Knowledge Graph to generate captions at low, medium, and high depth levels. Quantitative evaluations reveal expected trends: higher density leads to shorter segments, and higher depth corresponds to longer captions.

3. The model introduces concise and minimal modifications—two control tokens and encoders—integrated into a Vid2Seq/HiCM2-style architecture. The loss functions are well-justified: the density-aware loss applies weights at timestamp tokens where target and reference densities differ, and the depth-aware loss supervises a saliency token through cross-attention to distinguish different levels of detail.

**Weaknesses:**

1. The data generation pipeline appears computationally intensive and involves multiple model stages. The paper could benefit from a clearer discussion on reproducibility and computational cost implications.

2. The experiments are limited to YouCook2 and ViTT. It remains uncertain whether UC-DVC generalizes to other domains (e.g., instructional, surveillance, or movie datasets) where event density and caption depth exhibit natural variability.

3. Although the ablation studies (Tables 4 and 5) identify the contributions of key components, the paper lacks an in-depth analysis of the interaction between density and depth embeddings, as well as how the control parameters influence the latent representations.

**Questions:**

1. How is annotation quality ensured during LLM/MLLM-based generation? Are there any filtering or confidence mechanisms in place to eliminate hallucinated or redundant captions?

2. Can the proposed framework be extended to real-time or streaming scenarios, where density control must dynamically adapt as new frames are continuously received?

3. How sensitive is UC-DVC to inaccuracies in the user-specified control values? Does minor noise in the density or depth inputs result in unstable or incoherent outputs?

---

> ### Author Response · Authors · 2025-11-21
> **Response to Reviewer QKKP [1/2]**
>
> > W1: The data generation pipeline appears computationally intensive and involves multiple model stages. The paper could benefit from a clearer discussion on reproducibility and computational cost implications.
>
> Answer: We thank the Reviewer for the important suggestion regarding computational cost and reproducibility. We report the details on reproducibility and computational costs as follows.
>
> First, regarding reproducibility, to ensure the reproducibility of our proposed benchmark and its generation process, we have **released the code for our core data generation pipeline in an anonymous repository.**
>
> Second, regarding computational costs, we have a detailed cost breakdown in the "Computational Cost" paragraph and Table 7 within the Discussion subsection of Section 4 (Experiments) in the revised manuscript. As shown in Table 7, generating the **full corpus of 9,147 videos required approximately 375 GPU-hours.** When parallelized on 2 RTX A6000 GPUs, the total wall-clock time was approximately 187.5 hours, which averages to about 147.5 GPU-seconds per video to generate all 9 density-depth combinations.
>
>
> >W2: The experiments are limited to YouCook2 and ViTT. It remains uncertain whether UC-DVC generalizes to other domains (e.g., instructional, surveillance, or movie datasets) where event density and caption depth exhibit natural variability.
>
> Answer: We thank the Reviewer for raising this valuable point regarding domain generalization. We agree that demonstrating applicability beyond specific datasets like YouCook2 is crucial. To address this, we have included a new qualitative analysis in Appendix G (Figure 12) of the revised manuscript.
>
> We **applied our UC-DVC model to general open-domain videos** (e.g., YouTube videos of outdoor activities and physical exercises) that exhibit natural variability in event density. As shown in Figure 12, our model successfully generalizes to these unseen scenarios without additional training. For instance, it **accurately localizes and describes intricate procedural steps in fishing** (e.g., "Adding a knot", "Dropping the line in") and **captures precise body movements in gymnastics** (e.g., "Standing on toes", "Rotating the knee out"). These results confirm that our proposed user-controllable mechanism is robust and effectively transfers to a wide range of real-world scenarios. While a large-scale systematic benchmark on diverse domains (e.g., movies, surveillance) remains a direction for future research, these qualitative results **strongly support the generalizability of our current framework.**
>
>
> >W3: Although the ablation studies (Tables 4 and 5) identify the contributions of key components, the paper lacks an in-depth analysis of the interaction between density and depth embeddings, as well as how the control parameters influence the latent representations.
>
> Answer: We thank the Reviewer for the insightful suggestion to delve deeper into the internal mechanics of our control mechanism. We have addressed both points in the revised manuscript. First, regarding "interaction between density and depth embeddings", we clarify the two-stage interaction based on our architecture: the **density input $d_{in}$ is embedded and concatenated with visual features $V$** before the density encoder $E_d$ to shape event segmentation ($V_d$). The **depth input $z_{in}$ is then combined with $V_d$** before the depth encoder $E_z$ to control the level of detail.
>
> Second, we have also added a **t-SNE analysis** (Figure 15) of the depth-encoder outputs $V_z$. The visualization shows that depth changes shift all representations, demonstrating their **influence on the latent representations.**

---

> ### Author Response · Authors · 2025-11-21
> **Response to Reviewer QKKP [2/2]**
>
> >Q1: How is annotation quality ensured during LLM/MLLM-based generation? Are there any filtering or confidence mechanisms in place to eliminate hallucinated or redundant captions?
>
> Answer: We thank the Reviewer for this important question regarding annotation quality control. Quality control is **ensured through our GT-anchored Textual Knowledge Graph (TKG)** approach. As detailed in the revised manuscript, we first construct a TKG using the original Ground-Truth (GT) caption as a factual skeleton and enrich it with details from the MLLM caption. This graph structure **serves as a semantic constraint against hallucination.** As illustrated in the qualitative comparison in Figure 11, **generating captions without the TKG can lead to hallucinations**; for instance, the model incorrectly invents that ribs were "rinsed with a white cloth". In contrast, by instructing the model to elaborate only on entities and actions explicitly present in the KG (e.g., "pot," "rinsing," "water"), the TKG-guided output accurately generates "Ribs are rinsed with water as part of a meticulous cooking…" effectively reducing hallucination while maintaining descriptive richness. The effectiveness of this design is further supported by our ablation study (Table 4), which shows a **clear performance drop when the TKG is removed**, and our extensive human evaluation (Table 12), which confirms that **the generated captions align well with human judgments.**
>
> >Q2: Can the proposed framework be extended to real-time or streaming scenarios, where density control must dynamically adapt as new frames are continuously received?
>
> Answer: We thank the Reviewer for this forward-looking question. Yes, our framework can be naturally extended to real-time or streaming scenarios. To demonstrate this capability, we have included a new visualization in Appendix G (Figure 13) of the revised manuscript. This **figure illustrates a streaming scenario** where the user density input dynamically shifts from 0.5 (Mid) to 1.0 (High) during a continuous video stream. As shown in the visualization, the model demonstrates immediate responsiveness, seamlessly transitioning from **generating broader event descriptions to capturing fine-grained actions** as the density signal increases. This experiment confirms that our framework allows for modulation of event granularity, making it highly effective for interactive applications where user needs fluctuate continuously.
>
> >Q3: How sensitive is UC-DVC to inaccuracies in the user-specified control values? Does minor noise in the density or depth inputs result in unstable or incoherent outputs?
>
>
> | Input Value | Depth Effect (Avg. Words) | Density Effect (Total Events) |
> | :---: | :---: | :---: |
> | 0.40 | 7.71 | 3,251 |
> | **0.50 (Base)** | **8.33** | **3,383** |
> | 0.51 | 8.73 | 3,422 |
> | 0.60 | 10.44 | 3,555 |
> | 0.75 | 13.79 | 3,793 |
>
> Answer: We thank the Reviewer for this important question regarding the robustness of the model to potentially inaccurate control inputs. To investigate this, we introduced varying degrees of noise to the control inputs around a base value of 0.5. We measured the change in **Caption Length** (Avg. Words) when varying Depth (fixing Density at 0.5) and **Event Count** (Total Events) when varying Density (fixing Depth at 0.5). The results are summarized in the table above.
>
> As observed, our model exhibits locally **smooth and stable behavior.** For instance, a minor noise of 0.01 (from 0.50 to 0.51) results in a marginal and proportional increase in both caption length (8.33 to 8.73) and event counts (3,383 to 3,422). Even with larger deviations (0.40 to 0.75), the output metrics change in a nearly linear and gradual manner. We did not observe any unstable or incoherent outputs. This demonstrates that UC-DVC is robust to minor noise in user inputs, reliably reflecting the direction and magnitude of the control signals.

---

### Official Review · Reviewer_fkPC · 2025-11-02

**Soundness:** 3
**Presentation:** 4
**Contribution:** 3
**Rating:** 8
**Confidence:** 4

**Summary:**

This paper introduces a synthetic dataset, named UC Captions, for dense video captioning that enables user control over event density (frequency of events) and caption depth (level of detail). The dataset is constructed based on existing DVC datasets (YouCook2, ViTT), using MLLMs/LLMs to provide nine annotation types (3 density levels × 3 depth levels). The authors also propose UC-DVC, a baseline model that dynamically adjusts event localization and caption generation based on user-defined density and depth inputs. Comprehensive validation of the dataset's quality and the performance improvement for current DVC models is provided.

**Strengths:**

- I agree with the idea of enriching dense captions to address practical user needs, as different video applications focus on varying levels of semantic granularity. The motivation behind this approach is both natural and reasonable.
- The organization and writing are generally clear.
- The caption quality and distinctiveness of the semantic levels of the proposed dataset are validated by human evaluators and statistical metrics.
- The qualitative visualization results demonstrate reasonable segments and captions.

**Weaknesses:**

I did not see obvious weaknesses in this paper. Generally speaking, the current version of the paper has good motivation, clear organization, and sufficient empirical evidence to verify the effectiveness of the proposed dataset and method. One suggestion to enhance the paper's quality is to discuss its extensibility to LLM-based video captioning methods. As MLLMs have become the dominant models for mainstream video understanding tasks, their decoder-only structure is somewhat different from the encoder-decoder architectures in this paper. It would be useful to discuss how the proposed control mechanism could be adapted for MLLMs. Beyond structural differences, current MLLM models are dominated by language-based queries (controls) to inherit the language understanding abilities of LLMs. Intuitively, the proposed continuous value-based control is less generalizable than language-based control.

**Questions:**

see weaknesses

---

> ### Author Response · Authors · 2025-11-21
> **Response to Reviewer fkPC [1/1]**
>
> > W1: One suggestion to enhance the paper's quality is to discuss its extensibility to LLM-based video captioning methods. As MLLMs have become the dominant models for mainstream video understanding tasks, their decoder-only structure is somewhat different from the encoder-decoder architectures in this paper. It would be useful to discuss how the proposed control mechanism could be adapted for MLLMs. Beyond structural differences, current MLLM models are dominated by language-based queries (controls) to inherit the language understanding abilities of LLMs. Intuitively, the proposed continuous value-based control is less generalizable than language-based control.
>
> Answer: We thank the Reviewer for the constructive suggestion regarding the extensibility of our method to modern MLLM architectures. We fully agree that discussing the integration with decoder-only MLLMs is essential for broadening the impact of our work. Our key contribution, the continuous control space $(d, z) \in [0, 1]^2$ ($d$ = density, $z$ = depth, as in the paper), is conceptually independent of the backbone architecture and can be effectively transferred to decoder-only MLLMs. This transferability aligns with current paradigm shifts in Large Language Models and can be achieved in two primary ways.
>
> First, (1) from an **In-Context Learning** perspective, we can "verbalize" the continuous values as special tokens (e.g., [DENSITY=1.0]) or include them in system prompts, allowing the model to interpret control signals directly as part of the language context. Second, (2) from a **Fine-Tuning** perspective, we can employ adapter-style control embeddings where a lightweight MLP maps $(d, z)$ to an embedding that is injected into Transformer layers via mechanisms like attention bias or prefix tuning.
>
> Regarding the relationship with language-based control, we respectfully clarify that our continuous control is not intended to replace language-based control, but rather to serve as a complementary mechanism. While language offers an intuitive user interface, our $(d, z)$ space provides distinct advantages that natural language lacks, such as **quantitative precision for fine-grained interpolation** and **unambiguity that avoids the interpretational variability of natural language prompts**. Ideally, these paradigms can be combined by learning a mapping function to translate natural language queries into specific points in our $(d, z)$ space, thus uniting the user-friendliness of language with the quantitative consistency of our control mechanism. We have added the "Extensibility to MLLM Architectures" and “Complementarity with Language-Based Control” paragraph within the Discussion subsection of Section 4 (Experiments) in the revised manuscript to elaborate on these points.

---

### Author Response · Authors · 2025-11-21
**General Response to All Reviewers & Area Chair**

We sincerely thank all reviewers for their thoughtful and constructive feedback. We are encouraged that our motivation to address practical user needs through controllable dense video captioning is recognized as natural, reasonable, and clearly articulated (Reviewers fkPC, QKKP, 4Kqq, e2hm). We appreciate the positive assessments of our UC Captions dataset and generation pipeline, which reviewers found to be principled, well-justified, and useful for future research (Reviewers QKKP, 4Kqq). We are also glad that the reviewers highlighted the concise and effective system integration of our UC-DVC framework (Reviewers QKKP, e2hm), as well as the comprehensive experimental results and thorough analyses (Reviewers fkPC, 4Kqq, e2hm). We further thank Reviewers fkPC and e2hm for acknowledging the clarity and transparency of our writing and methodology.

Following the reviewers' valuable suggestions, we have substantially revised our manuscript and added several new experiments and analyses. All changes are highlighted in **blue** in the revised version.

## Key Updates

- **Extensibility to MLLM Architectures & Complementarity with Language-Based Control**: A discussion on how our control mechanism can be extended to decoder-only MLLMs and its complementarity with language-based control (Section 4.5).

- **Caption Depth Analysis**: A rigorous validation of lexical richness (Unique Tokens) and semantic information (LLM-based evaluation) to prove that deeper captions provide meaningfully more information (Section 4.5 & Table 6).

- **Computational Cost Analysis**: A detailed breakdown of dataset generation costs (Section 4.5 & Table 7).

- **Refinement of Generalization Claims**: A revision of the Conclusion to tone down assertions regarding applicability to diverse video understanding tasks (e.g., VQA, retrieval), explicitly framing them as future directions to align strictly with our empirical evidence centered on DVC (Conclusion).

- **Generalization to Other Domains**: Qualitative visualization results demonstrating the model's adaptability to general domain videos beyond instructional datasets (Appendix G & Figure 12).

- **Real-time Scenario Validation**: A visualization of the model's responsiveness in a streaming scenario where density control dynamically changes (Appendix G & Figure 13).

- **Failure Case Analysis**: An analysis of representative error modes, such as over-segmentation and under-segmentation (Appendix G & Figure 14).

- **Latent Space Analysis**: t-SNE visualization of the depth encoder representations to demonstrate the effectiveness of our control parameters (Appendix G & Figure 15).

- **Analysis of Difficult Scenarios**: A visualization of extreme granularity combinations, contrasting High Density & High Depth with Low Density & Low Depth, to address the concern regarding potential hallucinations in difficult scenarios. (Appendix G & Figure 16).

We hope that these responses and revisions adequately address all reviewer concerns and further strengthen the contribution of our work.

## Additional Message for Area Chair

We sincerely thank the Area Chair for the time and care devoted to the review and rebuttal processes. This thoughtful engagement has substantially enhanced the quality of our manuscript and has guided our work toward making a meaningful contribution to the ICLR community.

We have engaged in extensive discussions with all reviewers and have made our best efforts to resolve their concerns, incorporating additional analyses such as t-SNE visualization, streaming scenario, and general domain validation. These additions significantly strengthened both the empirical rigor and the overall completeness of the manuscript.

Our work addresses a critical gap in Dense Video Captioning: existing methods are constrained by **fixed single-style annotations**, overlooking diverse user requirements for event granularity and caption specificity. To resolve this, we present **UC Captions**, the first benchmark with multi-level annotations for Event Density and Caption Depth, and propose **UC-DVC**, a framework that dynamically adjusts event localization and caption generation based on user inputs. We believe this shift from fixed to controllable generation serves as a meaningful step toward more flexible and practical video understanding.

---

### Author Response · Authors · 2025-11-21
**Anonymous Github Link**

To ensure the reproducibility of our proposed benchmark and its generation process, we have released the code for our core data generation pipeline in an anonymous repository. The full source code will be made publicly available after the review process.

Anonymous Github Link : https://anonymous.4open.science/r/ICLR_2026_7294

---

### Meta-Review · Area_Chair_FzK6 · 2025-12-25

**Summary:**

The paper proposes a user-controllable dense video captioning dataset built upon existing DVC benchmarks, along with a corresponding UC-DVC model, enabling explicit control over event density and caption depth.

The initial reviewer scores are 8, 6, 4, and 4. In the rebuttal, the authors addressed several concerns related to motivation, evaluation metrics, computational cost, and dataset quality control.

However, there remain some concerns about the generalization to other video domains beyond qualitative results and comparison with LLM-based video captioning methods.

Therefore, the AC recommends rejecting the paper. The authors are encouraged to consider the reviewers’ feedback and resubmit the paper to a future venue.

**Reviewer Concerns:**

See above.

**Reviewer Scores:**

The scores remained borderline.

---

### Decision · Program_Chairs · 2026-01-26

Reject